# Essential angiosperm-specific subunits of HDA19 histone deacetylase complexes in Arabidopsis

Na Liu [1,2], Jia-Xin Li[2], Dan-Yang Yuan [2], Yin-Na Su[2], Pei Zhang[3], Qi Wang[2], Xiao-Min Su[2], Lin Li[2], Haitao Li [3], She Chen[2] & Xin-Jian He [2,4 ✉]

## Abstract

Although the *Arabidopsis thaliana* RPD3-type histone deacetylase HDA19 and its close homolog HDA6 participate in SIN3-type histone deacetylase complexes, they display distinct biological roles, with the reason for these differences being poorly understood. This study identifies three angiosperm-specific HDA19-interacting homologous proteins, termed HDIP1, HDIP2, and HDIP3 (HDIP1/2/3). These proteins interact with HDA19 and other conserved histone deacetylase complex components, leading to the formation of HDA19-containing SIN3-type complexes, while they are not involved in the formation of HDA6-containing complexes. While mutants of conserved SIN3-type complex components show phenotypes divergent from the *hda19* mutant, the *hdip1/2/3* mutant closely phenocopies the *hda19* mutant with respect to development, abscisic acid response, and drought stress tolerance. Genomic and transcriptomic analyses indicate that HDIP1/2/3 and HDA19 co-occupy chromatin and jointly repress gene transcription, especially for stress-related genes. An α-helix motif within HDIP1 has the capacity to bind to nucleosomes and architectural DNA, and is required for its function in Arabidopsis plants. These findings suggest that the angiosperm SIN3-type complexes have evolved to include additional subunits for the precise regulation of histone deacetylation and gene transcription.

**Keywords** Histone Deacetylation; HDA19; Transcriptional Repression; Development; Stress Response
**Subject Categories** Chromatin, Transcription & Genomics; Plant Biology

## Introduction

Histone acetylation is associated with active transcription and is dynamically regulated by histone acetylases (HATs) and histone deacetylases (HDACs) (Kuo and Allis, 1998). The dynamic regulation of histone acetylation plays a crucial role during plant development and in response to various environmental cues (Luo et al, 2017; Vafina and Stupak, 2023; Chen et al, 2024). HDACs are conserved in eukaryotes from yeast to mammals and plants (Pandey et al, 2002; Seto and Yoshida, 2014). As yet, a total of 18 HDACs in *Arabidopsis thaliana* are classified into three classes, including Reduced Potassium Dependence3/Histone Deacetylase-1(RPD3/HDA1), SIRTUIN2-like HDACs and plant-specific HDACs (Pandey et al, 2002). Two closely related Arabidopsis RPD3-type HDACs, HDA6 and HDA19, have been extensively studied, indicating that they are involved in hormone signaling, development phase transition, and stress responses, suggesting that they have redundant functions (Wu et al, 2000; Zhou et al, 2005; Tanaka et al, 2008; Chen and Wu, 2010; Chen et al, 2020).

In eukaryotes, the RPD3-type HDACs can form evolutionarily conserved SIN3-type histone deacetylase (HDAC) complexes, such as RPD3L and RPD3S complexes in yeast and SIN3A and SIN3B complexes in human (Florens et al, 2006; Chen et al, 2012; Adams et al, 2020). In Arabidopsis, HDA19 and HDA6 separately interact with the evolutionarily conserved SIN3-like (SNL) proteins (SNL1 to SNL6), HDC1/RXT3, and MSI1 to form SIN3-type HDAC complexes (Hennig et al, 2003; Alexandre et al, 2009; Perrella et al, 2013; Wang et al, 2013, 2016; Mehdi et al, 2016; Ning et al, 2019; Xu et al, 2022). These complexes are analogous to the yeast RPD3L and RPD3S complexes and the human SIN3A and SIN3B complexes (Ning et al, 2019; Adams et al, 2020). The conserved components within the Arabidopsis SIN3-type HDAC complexes are implicated in the regulation of abscisic acid (ABA) signaling (Perrella et al, 2013; Wang et al, 2013; Mehdi et al, 2016). SNL1 and SNL2 were reported to influence ABA, ethylene, and auxin signaling, thereby regulating seed dormancy and plant growth (Wang et al, 2013, 2016). HDC1, analogous to the yeast RPD3L complex subunit Rxt3, was found to exhibit a negative effect on ABA signaling and serve as a rate-limiting regulator of plant growth (Perrella et al, 2013). MSI1 was demonstrated to mediate histone deacetylation and transcriptional repression at ABA-responsive gene and thereby fine-tune ABA signaling (Mehdi et al, 2016). Similarly, HDA6 and HDA19 are also involved in the regulation of ABA signaling (Chen and Wu, 2010; Chen et al, 2010; Mehdi et al, 2016), aligning with the finding that SNLs, HDC1, and MSI1 are components of both HDA6- and HDA19-containing SIN3-type HDAC complexes (Perrella et al, 2013; Ning et al, 2019).

However, HDA19 and HDA6 are also involved in specialized roles. HDA19 can regulate embryogenesis (Zhou et al, 2013b), seed development and maturation (Gao et al, 2004, 2015; Wang et al,

[1]College of Life Sciences, Beijing Normal University, Beijing, China. [2]National Institute of Biological Sciences, Beijing, China. [3]School of Basic Medical Sciences, Tsinghua University, Beijing, China. [4]Tsinghua Institute of Multidisciplinary Biomedical Research, Tsinghua University, Beijing, China. ✉E-mail: hexinjian@nibs.ac.cn

2013), shoot regeneration (Temman et al, 2023), phytochrome signaling (Guo et al, 2023), and pathogen response (Kim et al, 2008; Choi et al, 2012). Disruption of HDA19 causes severe developmental defects that are not found in the mutant of HDA6 (Tian et al, 2003; Ning et al, 2019), indicating that HDA19 and HDA6 have independent functions. Unlike HDA19, HDA6 contributes to DNA methylation, transposon and transgene silencing, and heterochromatin condensation (Aufsatz et al, 2002; Earley et al, 2006; He et al, 2009; Tessadori et al, 2009; To et al, 2011; Liu et al, 2012; Blevins et al, 2014; Yu et al, 2017; Feng et al, 2021). HDA6 can physically interact with the DNA methyltransferase MET1 and the histone methyltransferases SUVH4/5/6, thereby regulating histone deacetylation and heterochromatin silencing (To et al, 2011; Liu et al, 2012; Yu et al, 2017). In terms of flowering time, HDA6 represses the expression of *FLC*, a core flowering repressor gene, to promote flowering (Wu et al, 2008; Yu et al, 2011; Xu et al, 2022), which is different from the mechanisms underlying the regulation of flowering time by HDA19 (Ning et al, 2019). Nevertheless, the mechanisms underlying the differential functional specificities between HDA19 and HDA6 have yet to be elucidated.

Here, we identified three previously uncharacterized homologous HDA19-interacting proteins (HDIP1/2/3) using affinity purification combined with mass spectrometry (AP-MS). We found that HDIP1/2/3 interact with HDA19, SNLs, HDC1, and MSI1, but not with HDA6, thus specifically contributing to the assembly of HDA19-containing histone deacetylase complexes in Arabidopsis. HDIP1/2/3 display redundant functions, and the loss-of-function mutants of HDIP1/2/3 and HDA19 exhibit highly similar defects in both plant development and ABA sensitivity. Additionally, we demonstrated that HDIP1/2/3 and HDA19 co-localize on chromatin and co-regulate transcriptional repression via both histone deacetylation-dependent and -independent mechanisms. We discovered that an α-helix motif in HDIP1 binds to the nucleosome and the four-way junction DNA, which mimics the DNA entry and enter sites of the nucleosome, and this binding is essential for the function of HDIP1 in Arabidopsis plants. These findings have uncovered previously unknown histone deacetylase complex subunits and highlighted an angiosperm-specific feature of the SIN3-type histone deacetylase complex, which is crucial for plant development and stress responses.

## Results

### Identification of plant-specific subunits of HDA19-containing HDAC complexes

The closely related RPD3-type HDACs HDA6 and HDA19 have been found to interact with conserved HDAC complex components, such as SIN3-like proteins (SNLs), HDC1, and MSI1, thereby forming conserved SIN3-type HDAC complexes in *Arabidopsis thaliana* (Ning et al, 2019). While HDA6 is known to play a role in DNA methylation and heterochromatin silencing (Aufsatz et al, 2002; He et al, 2009; To et al, 2011; Liu et al, 2012; Yu et al, 2017; Feng et al, 2021), HDA19 is predominantly associated with plant growth and development (Tian et al, 2003; Zhou et al, 2013b; Ning et al, 2019). However, the molecular mechanisms underlying the functional specificities of HDA19 compared to HDA6 are not yet fully understood. We therefore

performed affinity purification followed by mass spectrometry (AP-MS) in transgenic plants expressing HDA19, HDA6, HDC1, MSI1, SNL3, SNL4, SNL5, and SNL6, which were tagged with Flag or Myc epitopes, to compare the compositions of the HDA6- and HDA19-associated HDAC complexes. Our AP-MS data indicated that three previously uncharacterized homologs (AT1G69360, AT1G26620, AT1G13940) were co-purified with HDA19, which we have designated these as HDA19-interacting proteins 1, 2, and 3 (HDIP1/2/3: HDIP1, HDIP2, and HDIP3), respectively (Fig. 1A). Notably, HDIP1/2/3 were also co-purified with conserved subunits of SIN3-type HDAC complexes, SNLs, HDC1, and MSI1, but not with HDA6 (Fig. 1A). Furthermore, we generated transgenic plants expressing HDIP1/2/3 with a Flag tag for AP-MS analysis, confirming that these proteins interact with the aforementioned HDAC subunits, with the exception of HDA6 (Fig. 1A). Like the genes encoding other HDAC subunits, the HDIP1/2/3 genes were widely expressed in various plant tissues, according to the Athena database (http://athena.proteomics.wzw.tum.de:5002/master_arabidopsisshiny/) (Appendix Fig. S1). These results suggest that HDIP1/2/3 serves as specific subunits within HDA19-containing HDAC complexes in Arabidopsis.

Given that HDIP1/2/3 had not been previously characterized, we conducted sequence alignments utilizing DNAMAN software and predicted their structures using AlphaFold. The results indicated that HDIP1/2/3 exhibit a high degree of similarity (Appendix Fig. S2A), with each containing four α-helices (α1, α2, α3, α4) (Fig. 1B; Appendix Fig. S2A,B). Furthermore, we identified a putative nuclear localization signal (NLS) within HDIP1/2/3 (Appendix Fig. S2A). To determine whether HDIP1/2/3 are conserved subunits of the HDAC complex across plants, we employed the Basic Local Alignment Search Tool (BLAST) to search for orthologues in the NCBI databases. Our findings indicate that orthologues of HDIP1/2/3 are exclusively found in angiosperms (Appendix Fig. S3), suggesting that these orthologues function as angiosperm-specific subunits of HDAC complexes.

### Determination of interactions within HDA19-containing HDAC complexes

To investigate how HDIP1/2/3 are incorporated into the HDAC complex, we performed Y2H and pull-down assays. These experiments were designed to map the interactions between HDIP1 and other HDAC subunits, using both full-length HDIP1 and various truncated constructs (Fig. 1C). The combined Y2H and pull-down assays revealed that the second and third α-helices (α2/3) of HDIP1 interact with SNL5 and SNL6 (Fig. 1C–E; Appendix Fig. S4B–D). In addition, we tested truncated versions of SNL5 and SNL6 in these assays and found that their conserved N-terminal paired amphipathic helix (PAH) domains are the key interaction sites with HDIP1 (Fig. 1C–E; Appendix Fig. S4C,D). Moreover, our results showed that HDIP1 interacts with HDA19 but not with HDC1 or MSI1 (Fig. EV1A–E; Appendix Fig. S4A); the C-terminal unstructured regions of HDIP1 and HDIP3 were identified as the segment responsible for this interaction (Figs. 1C–E and EV1A–C). Consistent with the previous finding of the interaction between HDA19 and SNL1 (Wang et al, 2013), our pull-down assay demonstrated the interaction between HDA19 and SNL5, and indicated that the N-terminal region of SNL5 is responsible for this interaction (Fig. 1D,E). Collectively, these results suggest that the second and third α-helices of

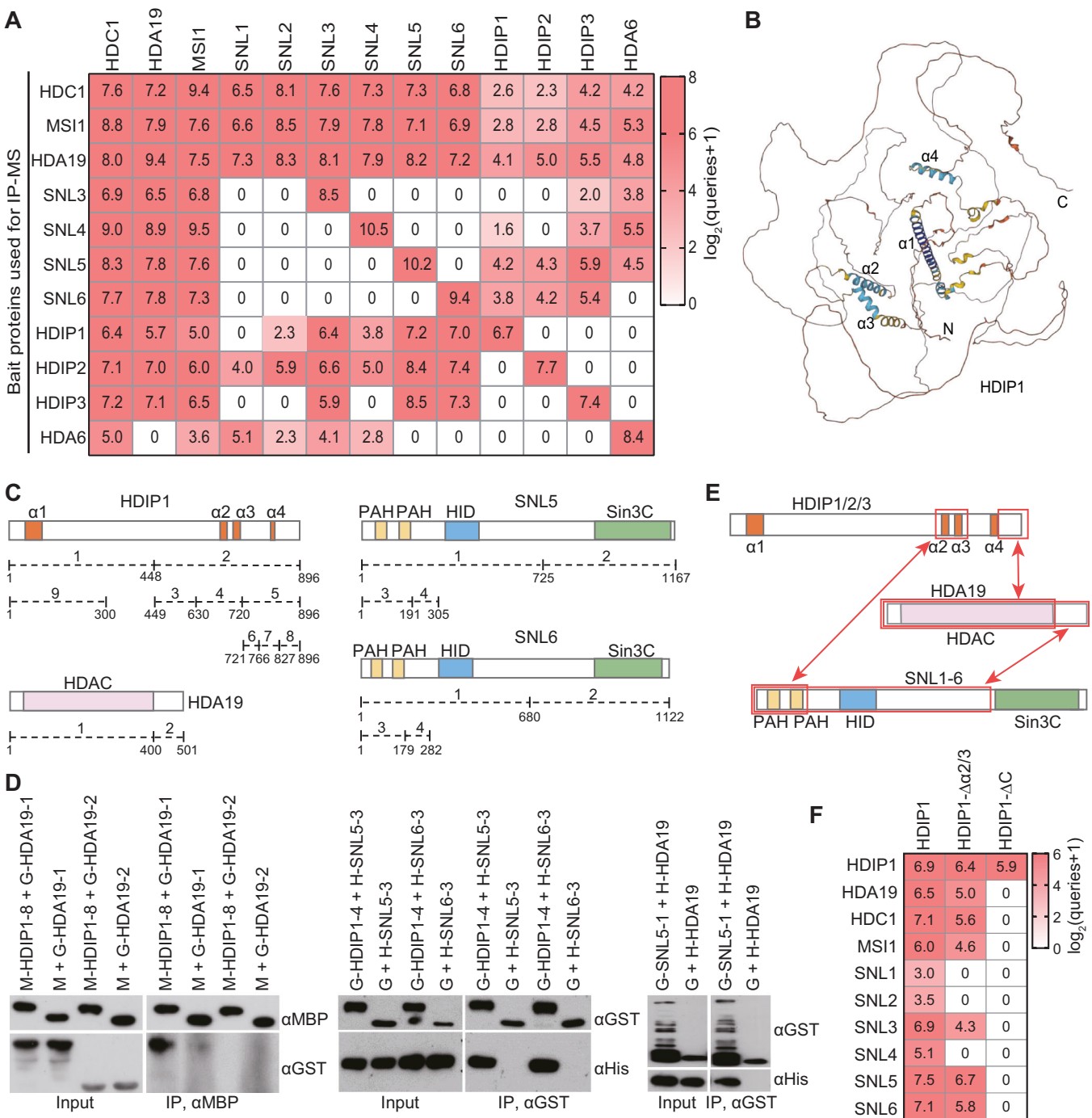

**Figure 1. Identification of previously uncharacterized subunits of HDA19-containing histone deacetylase complexes.**

(A) Heatmap showing proteins co-purified with histone deacetylase complex components as determined by AP-MS. The number represents the normalization of matched queries. (B) Prediction of the structure of HDIP1 by Alphafold. (C) Schematic diagrams showing full-length and truncated versions of HDIP1, HDA19, SNL5, and SNL6. The colorful boxes indicate conserved domains of proteins. (D) Determination of the in vitro interactions of HDIP1, SNL5/6, and HDA19 by pull-down assays. G GST, H His, M MBP. (E) Diagrams representing the interaction of HDIP1/2/3, SNLs, and HDA19 as detected by Y2H and pull-down assays. Red boxes indicate interacting regions. (F) Identification of proteins that interact with HDIP1-Flag, HDIP1-Δα2/3-Flag and HDIP1-ΔC-Flag in Arabidopsis plants by AP-MS. HDIP1-Δα2/3-Flag (Δ631–720 aa), deletion of the region of HDIP1 that interacts with SNLs. HDIP1-ΔC-Flag (Δ631–896 aa), deletion of the regions of HDIP1 that interact with SNLs and HDA19. The number represents the normalization of matched queries. Source data are available online for this figure.

HDIP1 interact with SNLs, while its C-terminal unstructured region interacts with HDA19.

To explore the roles of the second and third α-helices and the C-terminal unstructured region of HDIP1 in the assembly of the HDAC complex within Arabidopsis plants, we created transgenic plants expressing Flag-tagged HDIP1 variants. These variants lacked either the second and third α-helices (HDIP1-Δα2/3) or the C-terminal region (HDIP1-ΔC). AP-MS analysis indicated that the co-purification of HDAC components was disrupted by the absence of the C-terminal region but was only weakly affected by the removal of the second and third α-helices (Fig. 1F). Given that a putative NLS in the C-terminal region of HDIP1 was also deleted in HDIP1-ΔC (Fig. 1F; Appendix Fig. S2A), the disruption of co-purification of HDAC components may be attributed to the mislocalization of HDIP1-ΔC. However, our nuclear-cytoplasmic fractionation assay indicated that HDIP1 and HDIP3 were localized in the nucleus, and the deletion of the NLS did not affect the nuclear localization of HDIP1 (Appendix Fig. S5). These results suggest that the interaction of the C-terminal region of HDIP1 with HDA19 is essential for the incorporation of HDIP1 into the HDAC complex in Arabidopsis plants, while the interaction of the second and third α-helices of HDIP1 with SNLs seems to have an auxiliary role in its corporation into the complex.

## A similar effect of hdip1/2/3 and hda19 on plant development and gene expression

Previous studies have reported pleiotropic developmental defects in the hda19 mutant (Tian et al, 2003; Zhou et al, 2013b; Ning et al, 2019). To compare the functions of HDA19 and HDIP1/2/3 in Arabidopsis, we employed CRISPR-Cas9 system to generate various hdip mutants (Fig. EV2; Appendix Fig. S6). We found that while the hdip1, hdip2, and hdip3 single mutants did not exhibit visible defects compared to the wild-type, the hdip1/2/3 triple mutant and to a lesser extent the hdip1/2, hdip1/3, and hdip2/3 double mutants exhibited severe developmental defects, and the defects were highly similar to those observed in the hda19 mutant, which included enlarged cotyledons, curled rosette leaves, reduced leaf size, diminished plant height, increased primary branches, delayed flowering, and shorter siliques (Figs. 2A–G and EV2A–J). Notably, the developmental defects in the hdip1/2/3 mutant were significantly complemented by the HDIP1-Flag transgene (Appendix Fig. S7). In contrast, while the loss of other SIN3-type HDAC complex components also exhibits severe developmental phenotypes, these phenotypes are obviously different from those seen in the hda19 mutant (Ning et al, 2019). These findings suggest that within the HDA19-containing HDAC complex, HDIP1/2/3 are crucial for the functional specificities of HDA19 in the regulation of plant development.

To investigate the mechanisms by which HDIP1/2/3 and HDA19 collaborate in regulating development in Arabidopsis plants, we conducted RNA deep sequencing (RNA-seq) and determined differentially expressed genes (DEGs) in the hdip1/2/3 and hda19 mutants compared to the wild-type. Our analysis revealed a considerable number of DEGs ($\log_2$FC > 1 or < −1; $P$ value < 0.05) (Dataset EV1), specifically 1152 upregulated and 988 downregulated genes in the hdip1/2/3 mutants, and 1113 upregulated and 507 downregulated genes in the hda19 mutants (Fig. 3A). Notably, there was a significant overlap in the up- and

downregulated DEGs between the hdip1/2/3 and hda19 mutants (Fig. 3A). In addition, a scatter plot demonstrated a strong positive correlation between the effects of hdip1/2/3 and hda19 on gene expression (Fig. 3B), which is further corroborated by a heatmap (Fig. 3C). These analyses suggest that HDA19 and HDIP1 co-regulate gene expression across the entire genome. Gene Ontology (GO) analysis revealed that the upregulated DEGs in both hdip1/2/3 and hda19 mutants are related to biotic and abiotic stress responses, the abscisic acid (ABA) signaling pathway, and salicylic acid (SA)-mediated plant immunity (Fig. 3D). These included genes encoding components of the ABA receptors PYLs (Weiner et al, 2010), the ABA-responsive SnRK2 protein kinases (Fujii and Zhu, 2012), defense-responsive WRKY transcription factors (van Verk et al, 2011; Abbruscato et al, 2012; Hu et al, 2012; Chen et al, 2013; Gao et al, 2016; Wang et al, 2023b), and various elements of the SA signaling pathway (Zhang et al, 2010; Cui et al, 2017; Ullah et al, 2023) (Fig. 3E). Consequently, these findings suggest that HDIP1 and HDA19 work in concert to suppress the expression of a diverse set of stress-responsive genes.

## HDIP1/2/3 and HDA19 co-occupy chromatin and mediate transcriptional repression

To investigate the occupancy of HDIP1 and HDA19 at the whole-genome level, we performed ChIP-seq experiments using HDIP1-Flag and HDA19-Flag transgenic plants, respectively. Our ChIP-seq data indicated that HDIP1 and HDA19 showed a high positive correlation (Appendix Fig. S8A), and HDIP1 and HDA19 peaks were enriched in the promoter region and 5'-UTR of the genic region (Fig. 4A). We identified HDIP1 and HDA19 peaks annotated to 8951 and 13,366 genes, respectively, with the majority of them (7871, defined as group A) overlapping (Fig. 4B; Dataset EV2). Our analysis showed that the HDIP1 and HDA19 exhibited a similar distribution pattern over all HDA19 target genes, and were mainly located at the transcription start site (TSS)-flanking region (Fig. 4C; Appendix Fig. S8B,C). These data suggest that HDIP1 and HDA19 are associated with common target genes at the whole-genome level. To examine the potential role of HDIP1/2/3 in mediating the association of HDA19 with its target genes (group A), we conducted ChIP-seq for HDA19-Flag expressed in the hda19 and hda19hdip1/2/3 mutant backgrounds and found the enrichment level of HDA19 was not affected by the hdip1/2/3 mutation (Fig. EV3A,B). We also performed quantitative ChIP-PCR with a spike-in control from human cells, confirming that HDIP1/2/3 are dispensable for the association of HDA19 with chromatin (Fig. EV3C). Conversely, we sought to determine the effect of hda19 on the association of HDIP1 with chromatin using HDIP1-Flag transgenic plants in the hda19 mutant background. However, we found that while the transcript level of HDIP1-Flag transgene in the hda19 mutant was similar with that in the wild-type background (Fig. EV3D), the protein levels of the transgene were detected in the wild-type background but not in the hda19 mutant (Fig. EV3E), indicating that HDA19 is essential for maintaining that the protein level of HDIP1. This result supports the notion that HDIP1/2/3 are subunits of HDA19 histone deacetylase complexes in Arabidopsis plants.

We therefore investigated whether and how HDIP1/2/3 and HDA19 collaborate to regulate gene expression at their common target genes. A scatter plot showed that the expression levels of

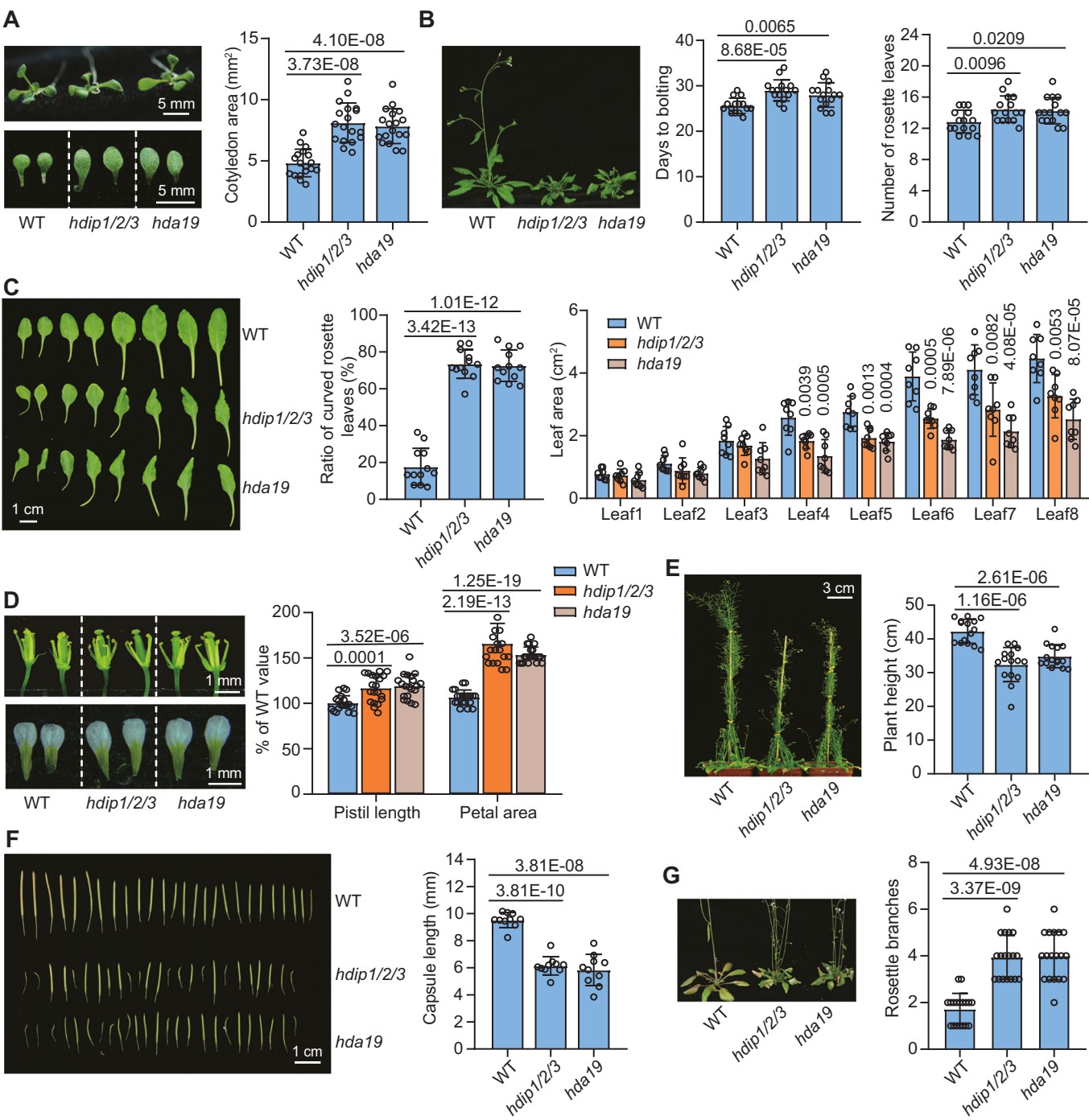

**Figure 2. Comparison of the phenotypes of the *hdip1/2/3* and *hda19* mutants.**

(A) Morphological phenotype of 10-day-old wild-type and mutant seedlings and the detached cotyledons from the seedlings. The cotyledon area was measured from 18 cotyledons for each genotype. (B) The flowering time in wild-type and mutants. The flowering time was determined as the days to bolting and the number of rosette leaves. Mean values and standard deviation (SD) are based on at least 15 plants. (C) Morphological characteristics of rosette leaves from 32-day-old plants. Images of rosette leaves (left), ratio of curved rosette leaves ($n = 12$, middle), and leaf area ($n = 8$, right) are presented. (D) Floral morphology of 38-day-old wild-type and mutant plants. Images of the flowers with sepals and petals removed (top) and separated petals (bottom) are shown. Statistical data indicate the percentage of pistil length and petal area in mutants compared to wild-type ($n = 20$, right). (E) Morphology and statistical analysis of plant height of 48-day-old plants in wild-type and the indicated mutants ($n = 15$). (F) Silique phenotype of wild-type and mutants. The morphology of siliques (left) and the statistical analysis of silique length (right) are shown. The length of siliques from ten plants was measured, with at least 15 siliques measured each. (G) Branching phenotypes of 45-day-old plants (left) and the number of primary rosette branches in wild-type and indicated mutants ($n = 18$, right). Data shown in (A–G) are means ± SD. *P* values were determined by two-tailed Student's *t* test. Source data are available online for this figure.

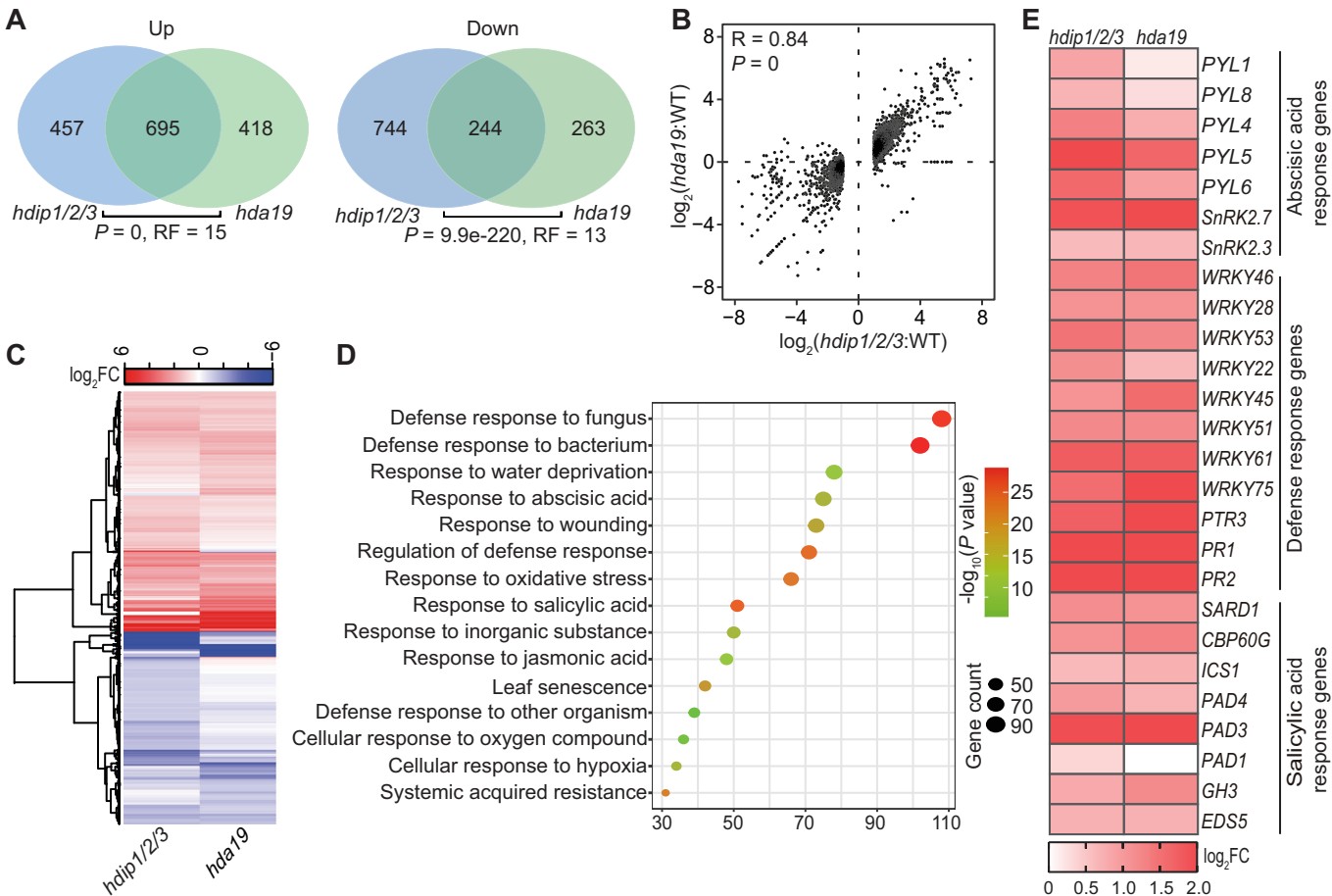

**Figure 3. HDIP1/2/3 and HDA19 co-regulate gene expression as determined by RNA-Seq.**

(A) Venn diagrams showing the overlap of upregulated and downregulated genes between the *hdip1/2/3* and *hda19* mutants (log$_2$FC > 1 or < −1, $P$ < 0.05). $P$ values were determined by the hypergeometric test (one-tailed). The representation factor (RF) is the number of observed overlapping genes divided by the expected overlapping number drawn from two independent groups. (B) The correlation of gene expression changes between *hdip1/2/3* and wild-type and between *hda19* and wild-type. $P$ values were determined by two-sided Pearson correlation test. (C) The expression changes of DEGs in the *hdip1/2/3* and *hda19* mutants relative to the type. Red and blue represent upregulated and downregulated genes, respectively. (D) GO analysis of co-upregulated genes identified in the *hdip1/2/3* and *hda19* mutants. The dot size and color indicate the number and enrichment of co-upregulated genes in each biological process, respectively. $P$ values were determined by one-sided Fisher's exact test. (E) The expression changes of representative upregulated genes in the indicated mutants relative to the wild-type.

HDIP1 and HDA19 shared target genes were prone to be upregulated in the *hdip1/2/3* and *hda19* mutants rather than be downregulated, and the up-regulation exhibits a high degree of positive correlation (Fig. 4D). As shown by Venn diagrams, the HDIP1 and HDA19 shared target genes were significantly overlapped with upregulated DEGs identified in *hdip1/2/3* and *hda19* mutants, while the overlaps between the shared target genes and downregulated DEGs in *hdip1/2/3* and *hda19* mutants were even lower than expected by chance (Fig. 4E), which supports the role of HDIP1/2/3 and HDA19 in mediating transcriptional repression. To further explore whether the chromatin binding levels of HDIP1 and HDA19 are associated with their impact on transcriptional repression, we individually assessed the enrichment of HDIP1 and HDA19 at three groups of genes, including HDIP1 and HDA19 shared target genes (group A), the overlap between group A genes and co-upregulated DEGs (defined as group B), and the overlap between group A genes and co-downregulated DEGs (defined as group C) (Fig. 4B,E). We found that the overall ChIP-

seq levels of both HDIP1 and HDA19 were significantly higher in the group B genes than in the group A and C genes (Fig. 4F). Gene ontology (GO) analysis indicated that the group B genes were enriched in terms related to abscisic acid (ABA) response, salicylic acid (SA) response, and defense response to bacteria (Appendix Fig. S9; Dataset EV3). These analyses suggest that HDIP1/2/3 and HDA19 cooperatively mediate transcription repression of genes responsive to biotic and abiotic stimuli.

## The histone deacetylation-dependent and -independent functions of HDIP1 and HDA19

Given that HDA19 is a histone deacetylase, we determined the histone deacetylation activity of HDA19 on free histone substrates using purified HDA19 from *HDA19-Flag* transgenic plants. Our in vitro histone deacetylation assay indicated that HDA19 can mediate histone deacetylation at all tested lysine sites (Fig. EV4A), suggesting that HDA19 can mediate histone deacetylation broadly

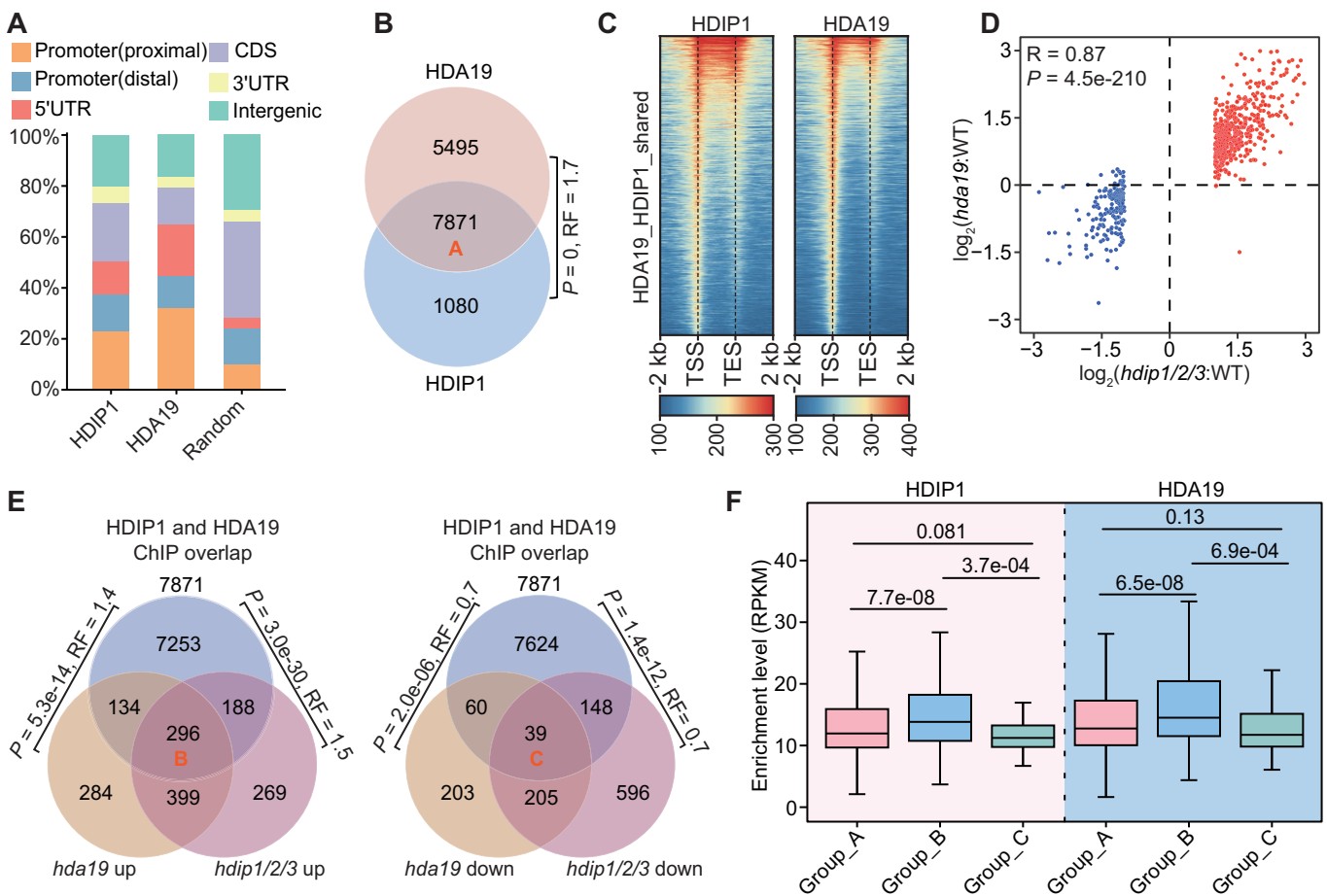

**Figure 4. HDIP1 and HDA19 co-occupy chromatin and mediate transcriptional repression.**

(A) Distribution of HDIP1 and HDA19 ChIP-seq peaks in genomic regions, including proximal promoter (1–400 bp upstream of the TSS), distal promoter (401–1000 bp upstream of the TSS), 5′ UTR, CDS, 3′ UTR, and intergenic regions. (B) The overlap between genes targeted by HDIP1 and genes targeted by HDA19. The overlapping genes are defined as group A genes. The *P* value was determined by the hypergeometric test (one-tailed). (C) Heatmaps showing the ChIP-seq signals of HDIP1 and HDA19 at their shared target genes. TSS, transcription start site; TES, transcription end site. The scale represents RPKM. (D) The correlation of the expression changes of the target genes shared by HDIP1 and HDA19 in the *hdip1/2/3* and *hda19* mutants relative to the wild-type. *P* values were determined by two-sided Pearson correlation test. (E) The overlap of HDIP1 and HDA19 co-occupied genes with genes that are up- or downregulated in the indicated mutants. The overlap of HDIP1 and HDA19 co-occupied genes with co-upregulated genes or co-downregulated genes in *hdip1/2/3* and *hda19* is categorized as group B (left) or group C (right), respectively. *P* values were determined by the hypergeometric test (one-tailed). The representation factor (RF) is the number of observed overlapping genes divided by the expected overlapping number drawn from two independent groups. (F) Box plot showing the ChIP-seq levels of HDIP1 and HDA19 at group A (*n* = 7871), group B (*n* = 296), and group C (*n* = 39) genes. In box plots, center lines and box edges are medians and the interquartile range (IQR), respectively. Whiskers extend within 1.5 times the IQR. *P* values were determined by the two-tailed Mann–Whitney *U* test (unpaired) for non-normally distributed data.

at multiple lysine sites. Considering that HDIP1/2/3 have been demonstrated to be components of HDA19-associated HDAC complexes, we inquired whether HDIP1/2/3 mediate HDA19-dependent histone deacetylation to promote transcriptional repression. Therefore, we conducted western blot analysis to determine the effect of *hdip1/2/3* and *hda19* on the global histone acetylation level, and found that the histone acetylation level was not affected in the *hdip1/2/3* or *hda19* mutants relative to the wild-type (Fig. EV4B). Consequently, we performed ChIP-seq to detect the effect of *hdip1/2/3* and *hda19* on histone H3 acetylation (H3Ac) at the whole-genome level. Consistent with the western blot results, the ChIP-seq analysis revealed that the overall H3Ac level was not significantly affected in the *hdip1/2/3* or *hda19* mutants compared to the wild-type (Fig. EV4C). Compared to the wild-type, our

ChIP-seq analysis identified 986 genes with increased H3Ac levels and 350 genes with reduced H3Ac levels in the *hdip1/2/3* mutant, as well as 1207 genes with increased H3Ac levels and 600 genes with reduced H3Ac levels in the *hda19* mutant (FC > 1.2 or <0.8, FDR < 0.05) (Fig. 5A; Dataset EV4), indicating that the genes with increased H3Ac level in both *hdip1/2/3* and *hda19* are more abundant than those with reduced H3Ac levels. Notably, a substantial portion of genes (731/1207) with increased H3Ac levels in the *hda19* mutant overlapped with those in the *hdip1/2/3* mutant (Fig. 5B). In addition, we found that the HDIP1 and HDA19 shared target genes comprise the majority of genes with increased levels of H3Ac (599/986 in *hdip1/2/3* and 767/1207 in *hda19*) and a small subset of genes with decreased levels of H3Ac (84/350 in *hdip1//2/3* and 233/600 in *hda19*) (Fig. 5C). These findings reveal that HDIP1/2/3 and

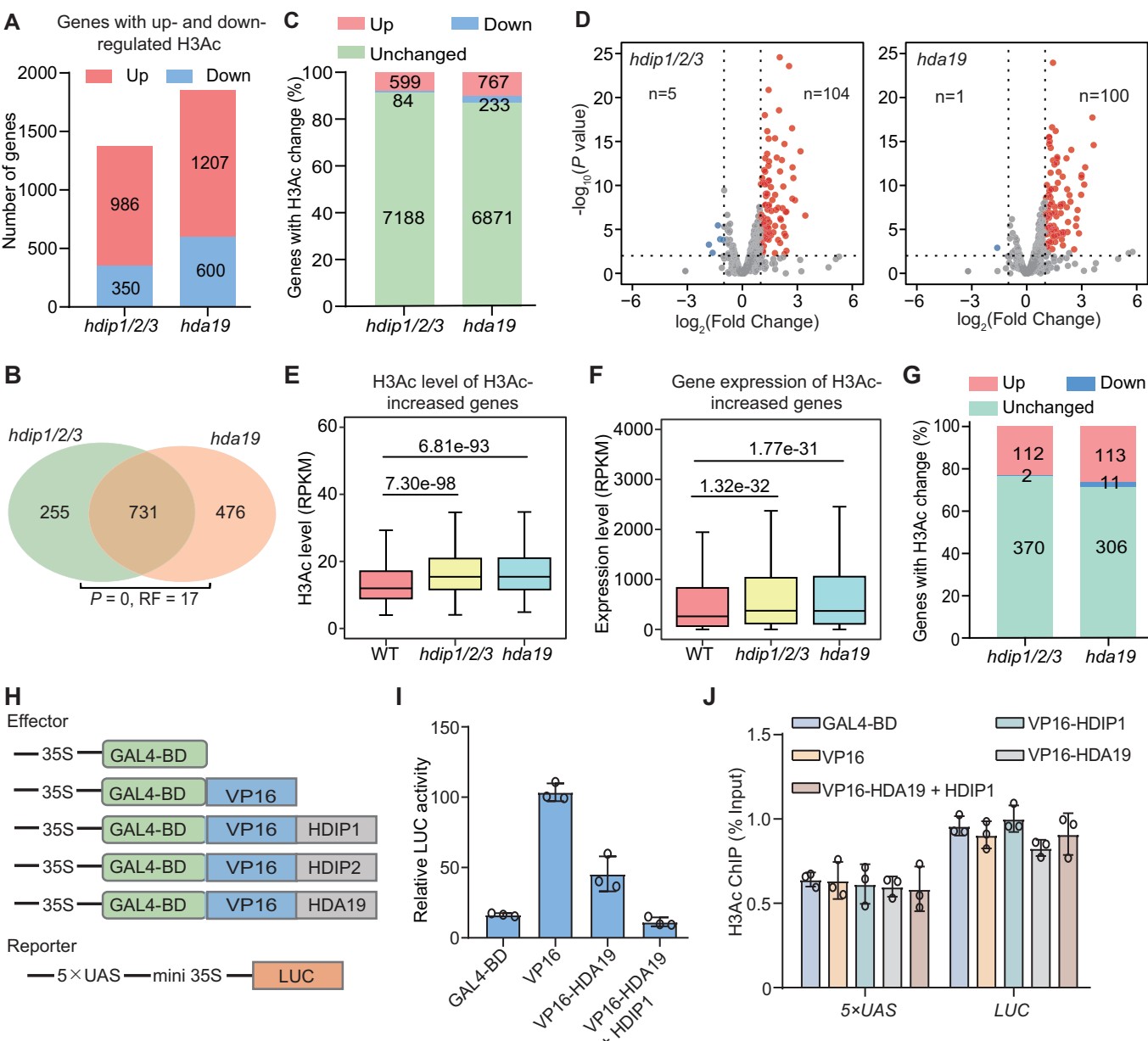

**Figure 5. HDIP1/2/3- and HDA19-mediated transcriptional repression depends on both histone deacetylation-dependent and -independent mechanisms.**

(A) Determination of genes with upregulated and downregulated H3Ac levels identified by ChIP-seq in *hdip1/2/3* and *hda19* mutants relative to the wild-type (FC > 1.2 or < 0.8, FDR < 0.05). Data are from two biological replicates. (B) The overlap of genes with increased H3Ac levels in *hdip1/2/3* and *hda19* mutants. The *P* value was determined by the hypergeometric test (one-tailed). (C) Percentage of HDIP1 and HDA19 co-occupied genes with upregulated, downregulated and unchanged H3Ac levels in *hdip1/2/3* and *hda19* mutants. Values are the number of genes. (D) The expression alterations of HDIP1 and HDA19 co-occupied genes with increased H3Ac in *hdip1/2/3* and *hda19* mutants. Red, blue, and gray dots represent genes showing upregulated, downregulated, and unchanged expression, respectively. *P* values were determined by the negative binomial distribution and Fisher's exact test. (E, F) Box plots showing the H3Ac level (E) and the expression level (F) of HDIP1 and HDA19 co-occupied genes with increased H3Ac levels in *hdip1/2/3* and *hda19* mutants. Sample size of each box plot in (E, F): HDIP1 and HDA19 co-occupied genes with increased H3Ac levels in *hdip1/2/3* mutant (*n* = 599). In box plots (E, F), center lines and box edges are medians and the interquartile range (IQR), respectively. Whiskers extend within 1.5 times the IQR. *P* values were determined by the two-tailed Mann–Whitney *U* test (unpaired) for non-normally distributed data. (G) Percentage of genes with upregulated, downregulated, and unchanged H3Ac levels in *hdip1/2/3* and *hda19* mutants. The HDIP1 and HDA19 co-occupied genes with upregulated expression in *hdip1/2/3* and *hda19* mutants were subjected to the analysis. Values are the number of genes. (H) Schematic representation of constructs used in the luciferase (LUC) reporter assay. The transcription factor VP16 fused with the DNA-binding domain of GAL4 (GAL4-BD) was used as a positive control. The expression of the luciferase reporter gene was driven by five repeats of the GAL4-binding upstream activation sequence (5×UAS) and the mini 35S. (I) The transcriptional repression ability of HDA19 combined with HDIP1 as detected by the *LUC* reporter assay. Values are means ± SD of three biological replicates. (J) The enrichment level of H3Ac at the *5×UAS* and *LUC* regions of the reporter gene. Quantitative ChIP-PCR was utilized to measure the H3Ac enrichment level, which was normalized to the input DNA. Values are means ± SD of three independent biological replicates. Source data are available online for this figure.

HDA19 jointly regulate histone deacetylation at a specific subset of their shared target genes.

To investigate whether HDIP1/2/3 and HDA19 repress transcription in a histone deacetylation-dependent manner, we examined the expression levels of HDIP1 and HDA19 shared target genes with increased H3Ac levels in the *hdip1/2/3* and *hda19* mutants. Volcano plots indicated that within the HDIP1 and HDA19 shared target genes showing increased H3Ac levels in the *hdip1/2/3* and *hda19* mutants, the genes with increased expression levels were markedly more abundant than those with decreased expression levels (Fig. 5D). In addition, the box plot indicated that at the HDIP1 and HDA19 shared target genes displaying an increased level of H3Ac in the *hdip1/2/3* mutant, the H3Ac level was also elevated in the *hda19* mutant (Fig. 5E). Moreover, the overall expression levels of these genes were significantly increased in *hdip1/2/3* as well as in *hda19* compared to the wild-type (Fig. 5F). These analyses suggest that the role of HDIP1/2/3 and HDA19 in mediating transcriptional repression is at least partially attributed to histone deacetylation.

Although HDIP1/2/3 and HDA19 were demonstrated to mediate histone deacetylation-dependent transcriptional repression, we also identified a number of HDIP1 and HDA19 shared target genes that exhibited increased expression levels without altering H3Ac levels in the *hdip1/2/3* and *hda19* mutants compared to the wild-type (Fig. 5G), suggesting that HDIP1/2/3 and HDA19 could also mediate transcriptional repression in a histone deacetylation-independent manner. Therefore, we tested whether HDIP1/2/3 and HDA19 are involved in transcriptional repression using a dual-luciferase (LUC) reporter system (Fig. 5H). While the effector construct expressing the transcription factor VP16 activated the LUC activity significantly, the fusion of HDIP1 or HDIP2 to VP16 substantially inhibited the activity (Fig. EV4D). We further fused a series of truncated versions of HDIP1 to VP16 and found that the truncated form HDIP1-2 exhibits a major repressive effect on transcription while the other truncated forms have minor effects (Fig. EV4E), confirming the role of HDIP1 in transcriptional repression. Moreover, we found that the fusion of HDA19 to VP16 could also suppress the role of VP16 in transcriptional activation (Fig. 5I). However, the effect of HDA19 on transcriptional repression was notably weaker than that of HDIP1 and HDIP2 (Figs. 5I and EV4D), suggesting that HDIP1/2 play a predominant role in transcriptional repression. Since we have demonstrated the interaction between HDIP1 and HDA19 within the HDAC complex, we introduced the over-expressed HDIP1 protein into the LUC reporter system along with the HDA19-VP16 fusion protein. We observed that while the fusion of HDA19 to VP16 had a moderate suppressive effect on the VP16-dependent LUC activity, the addition of the extra HDIP1 protein further reduced the LUC activity to the background level (Fig. 5I). To examine whether HDA19 and HDIP1 repress the expression of the *LUC* reporter genes through histone deacetylation, we performed quantitative ChIP-PCR using Arabidopsis mesophyll protoplast. The H3Ac levels of the *5×UAS* and *LUC* loci were not significantly affected by different effectors, despite the expression level of the reporter gene was affected (Fig. 5J). Together, these results suggest that HDIP1/2/3 have both histone deacetylation-dependent and -independent roles in mediating transcriptional repression.

## HDIP1 binds to a four-way junction DNA and nucleosomes

Given that the HDAC complex can mediate histone deacetylation on the nucleosome composed of a histone octamer wrapped by DNA, we performed an electrophoresis mobility shift assay (EMSA) to detect the potential DNA-binding ability of HDIP1. By using four truncated versions of the HDIP1 protein (HDIP1-a, HDIP1-b, HDIP1-c and HDIP1-d) (Fig. 6A), our EMSA results showed that HDIP1-c, but not the other truncated forms of HDIP1, is capable of binding to all the three approximately 200-bp double-stranded DNA probes (P1, P2, and P3), regardless of whether the DNA probes originate from an HDIP1-enriched genomic region (Figs. 6B,C and EV5A,B). To characterize the DNA-binding ability of HDIP1-c, we divided the probes P1 and P2 into four 50-bp fragments (Fig. EV5C,E), and then determined the binding of HDIP1-c to these 50-bp probes. Our EMSA results revealed that HDIP1-c can also bind to all the 50-bp DNA probes (Fig. EV5D,F), suggesting that HDIP1 binds to DNA in a sequence-independent manner. Interestingly, the affinity of HDIP1 for long DNA (200 bp) was stronger than that for short DNA (50 bp) (Fig. 6D and EV5D,F). By using increasing amounts of long DNA (200 bp) in EMSA, we found that the increase of long DNA (200 bp) along with a constant concentration of HDIP1 protein and short DNA (50 bp) led to the disappearance of the HDIP1-DNA (50 bp) band and the emergence of a higher-molecular weight band corresponding to HDIP1 bound to long DNA (200 bp) (Fig. 6E). When the concentration of long DNA was equal to that of short DNA, the HDIP1-DNA (50 bp) band was completely replaced by the HDIP1-DNA (200 bp) band (Fig. 6E). In addition, we purified the corresponding domain of HDIP3 and assessed its DNA-binding affinity using EMSA, which showed that HDIP3 also exhibited a stronger affinity for long DNA (200 bp) compared to short DNA (50 bp) (Fig. EV5G). These results suggest that the HDIP1/2/3 proteins have a preference for longer DNA.

Previous studies suggest that ~150 bp of DNA is the minimum DNA length required for the autonomous formation of DNA loop (Shore et al, 1981; Semsey et al, 2005). This gives rise to the possibility that HDIP1 might recognize a high-order structure of DNA rather than the linear DNA. Considering that the nucleosome is the substrate of the HDAC complex (Lee et al, 2021; Guan et al, 2023), HDIP1, as a subunit of the HDAC complex, is likely to bind to the architectural DNA within the nucleosome rather than free DNA. Compared to free DNA, the nucleosomal DNA forms a high-order structure and is likely bound by HDIP1. To test this hypothesis, we used four 50-nucleostide DNA strands to construct a four-way junction DNA that mimics the DNA structure in the nucleosome (Wang et al, 1998; Panday and Grove, 2016) (Fig. 6F), and then determined the binding of HDIP1 and HDIP3 to the DNA. Our EMSA results demonstrated that the DNA-binding region of HDIP1 and HDIP3 efficiently shifted the four-way junction DNA as well as the 200-bp double-stranded DNA but only partially shifted the same amount of 50-bp double-stranded DNA (Figs. 6G and EV5H), suggesting that the high-order DNA structure formed in both long DNA and four-way junction DNA can enhance the DNA-binding ability of HDIP1 and HDIP3. In addition, we assembled nucleosomes and conducted a nucleosome-binding assay using different truncated versions of HDIP1. We

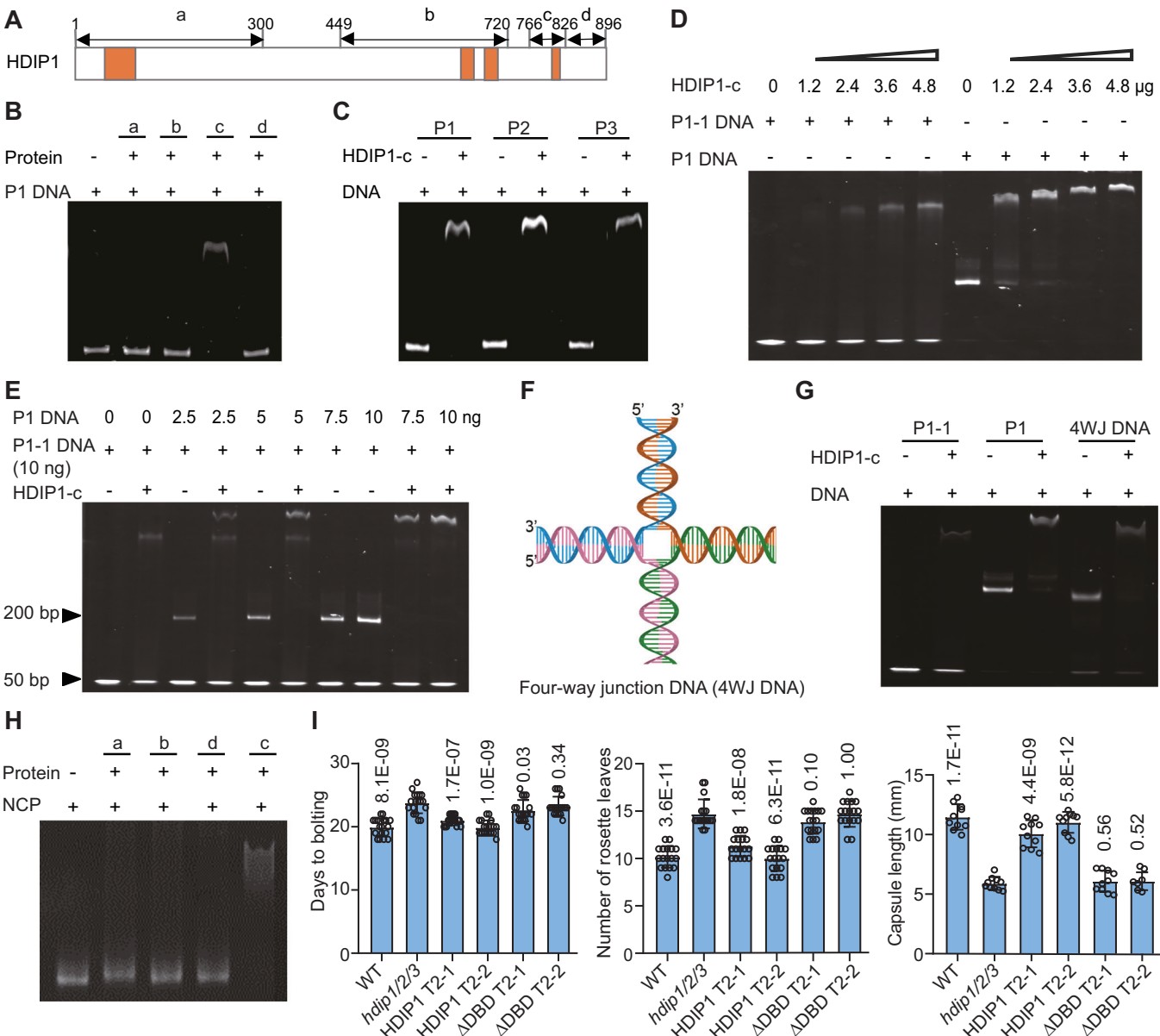

**Figure 6. An α-helix of HDIP1 binds to four-way junction DNA and nucleosomes and is required for the function of HDIP1 in Arabidopsis.**

(A) Diagrams showing truncated versions of the HDIP1 protein used in the DNA and nucleosome-binding assays. The truncated forms include HDIP1-a (1–300 aa), HDIP1-b (449–720 aa), HDIP1-c (766–826 aa), and HDIP1-d (826–896 aa). Orange boxes indicating the four α-helices within the HDIP1 protein. (B) Identification of the DNA-binding domain of HDIP1 by EMSA. P1 represents the 200-bp DNA probe from the HDIP1-enriched region of the *WRKY28* promoter. (C) Determination of the binding ability of HDIP1-c with the 200-bp P1, P2, and P3 probes by EMSA. P2 and P3 are from upstream and downstream of P1, and are not in the HDIP1-enriched genomic region. (D) The comparison of the binding ability of HDIP1-c with the 50-bp P1-1 probe and the 200-bp P1 probe. Increasing amounts of HDIP1-c protein were mixed with the same amount of P1 and P1-1 probes for EMSA. (E) The titration of the 200-bp P1 probe in EMSA. Increasing amounts of the 200-bp P1 probe were mixed with the constant amounts of the HDIP-c protein and the 50-bp P1-1 probe. (F) Schematic of four-way junction DNA (4WJ DNA) used in EMSA. (G) Determination of the binding ability of HDIP1-c with 4WJ DNA by EMSA. The 50-bp P1-1 probe and the 200-bp P1 probe were used as controls. (H) Determination of the binding ability of HDIP1 with nucleosomes. HDIP1-a, HDIP1-b, HDIP1-c, and HDIP1-d were used in the nucleosome-binding assay. NCP, nucleosome core particle. (I) Statistical analyses of days to bolting (*n* = 18), the number of rosette leaves (*n* = 17), and the length of siliques (*n* = 10, the average value of at least 15 siliques per plant) in the wild-type, *hdip1/2/3*, and *HDIP1-Flag* and *HDIP1-ΔDBD-Flag* transgenic plants in the *hdip1/2/3* mutant background. Data are means ± SD. *P* values were determined by two-tailed Student's *t* test for comparing the values between the *hdip1/2/3* mutant and the indicated genotypes. Source data are available online for this figure.

found that HDIP1-c also bound to nucleosomes while the other truncated versions of HDIP1 did not (Fig. 6H). These findings reveal that HDIP1 binds to DNA within nucleosomes, thereby facilitating histone deacetylation and transcriptional repression. To investigate whether the DNA-binding domain (DBD) of HDIP1

functions in Arabidopsis plants, we generated a construct expressing DBD-deleted HDIP1 (HDIP1-ΔDBD) and transformed it into the *hdip1/2/3* mutant. Compared to the wild-type *HDIP1* transgene that could complement the defects of the *hdip1/2/3* mutant in the regulation of flowering time and silique development,

the HDIP1-ΔDBD transgene was unable to complement the defects (Fig. 6I; Appendix Fig. S10A–C), suggesting that the DNA-binding ability of HDIP1 is essential for its function in Arabidopsis. To investigate whether the DNA-binding ability of HDIP1 is involved in its genome-wide association with chromatin in Arabidopsis, we performed ChIP-seq analysis on wild-type HDIP1 and HDIP1-ΔDBD, and found that the deletion of the DNA-binding domain (ΔDBD) did not affect the genome-wide association of HDIP1 with chromatin (Appendix Fig. S10D,E), suggesting that the DNA-binding domain is not required for the association of HDIP1 with chromatin. Our AP-MS analysis showed that the deletion of the DNA-binding domain (ΔDBD) did not disrupt the incorporation of HDIP1 into the HDA19-containing HDAC complex (Appendix Fig. S10F). In addition, our ChIP-seq data indicated that the hdip1/2/3 mutation did not affect the association of HDA19 with chromatin at the whole-genome level (Fig. EV3A–C), supporting the notion that the DNA-binding domain of HDIP1 takes effect after the HDA19 complex is associated with chromatin. Given that the DNA-binding domain is crucial for the biological function of HDIP1, we predict that its DNA-binding ability is crucial for facilitating the histone deacetylase activity of HDA19 within the HDAC complex.

## HDIP1/2/3 and HDA19 co-regulate the ABA response and drought tolerance

Considering that previous studies demonstrated that the hda19 mutation leads to increased sensitivity to ABA and the enhanced tolerance to drought stress (Chen and Wu, 2010; Ueda et al, 2018), we hypothesized that HDIP1/2/3 collaborated with HDA19 to suppress ABA signaling and drought stress tolerance. To test the hypothesis, we incubated the hdip1/2/3 and hda19 mutants and the wild-type on 1/2 MS medium with or without treatment by different concentrations of ABA, and observed their germination rates. Consistent with the previous report that the hda19 mutant is more sensitive to ABA than the wild-type (Chen and Wu, 2010; Ueda et al, 2018), our results indicated that while the germination rates were similar between the hda19 mutant and the wild-type under normal growth conditions, the germination rate was more significantly reduced by the ABA treatment in the hda19 mutant compared to the wild-type (Fig. 7A,B). Moreover, we found that the hdip1/2/3 mutant exhibited a similar ABA sensitivity phenotype as the hda19 mutant (Fig. 7A,B). ABA is a hormone that is involved in enhancing tolerance to various stress conditions, particularly drought (Yoshida et al, 2014). Therefore, we assessed the effect of hdip1/2/3 and hda19 on drought stress tolerance. Ten-day-old seedlings were grown in water-saturated soil and maintained without additional irrigation for 20 days. We found that a large majority (86.1% for the hdip1/2/3 mutant and 94.4% for the hda19 mutant) of the mutant plants survived after 2 days of rewatering, while only a small portion (5.5%) of the wild-type plants survived (Fig. 7C,D). Furthermore, we determined the effect of hdip1/2/3 and hda19 on the water loss of detached leaves and found that the rate of water loss was lower in the hdip1/2/3 and hda19 mutants than in the wild-type (Fig. 7E), suggesting that the hdip1/2/3 and hda19 mutants enhance drought stress tolerance by reducing water loss.

Notably, our RNA-seq data revealed that the expression levels of the ABA signaling pathway genes PYL1, PYL4, PYL5, PYL6, PYL8, SnRK2.3, and SnRK2.7 were enhanced in both the hdip1/2/3 and

hda19 mutants compared to the wild-type (Fig. 3E). In addition, our ChIP-seq results indicated that HDIP1 and HDA19 bind to the TSS-flanking regions of these ABA signaling pathway genes (Fig. 7F), corroborating the idea that HDIP1 and HDA19 directly suppress the transcription of these genes. Although the expression levels of these ABA signaling pathway genes are elevated in the hdip1/2/3 and hda19 mutants, the increased expression of most of these genes is not accompanied by the increased H3Ac levels (Fig. 7G; Appendix Fig. S11), suggesting that HDIP1 and HDA19 suppress the transcription of these genes primarily through a histone deacetylation-independent manner. Therefore, this study not only identifies previously unrecognized subunits of the HDA19-containing HDAC complex but also reveals the histone deacetylation-independent role of the complex in Arabidopsis plants.

## Discussion

In eukaryotes, the RPD3-type HDACs can form evolutionarily conserved SIN3-type HDAC complexes, such as RPD3L and RPD3S complexes in yeast and SIN3A and SIN3B complexes in mammals (Gregoretti et al, 2004; Yang and Seto, 2008; Venturelli et al, 2015). These complexes contain both conserved and species-specific subunits between yeast and mammals (Rundlett et al, 1996; Lewis et al, 2016). In Arabidopsis, previous studies have shown that the closely related RPD3-type HDACs HDA19 and HDA6 can separately interact with the conserved subunits of SIN3-type HDAC complexes, including SNLs, HDC1, and MSI1, resulting in the formation of the SIN3-type complexes (Perrella et al, 2013; Wang et al, 2013; Ning et al, 2019). However, the mechanisms underlying the functional specificities between HDA19 and HDA6 are yet to be fully understood. In this study, we identified that three uncharacterized homologs, HDIP1/2/3, form HDA19-containing SIN3-type HDAC complexes but not HDA6-containing ones in Arabidopsis. Although SNLs, shared subunits of HDA19 and HDA6 histone deacetylation complexes, can directly interact with HDIP1, as determined by in vitro pull-down assays, the interacting interface of SNLs is possibly hidden in the HDA6 complex, resulting in the absence of HDIPs in the HDA6 complex in Arabidopsis. Because HDIP1/2/3 orthologs are present in angiosperms but not in other plant species, these findings not only reveal the mechanisms underlying the functional specificities between HDA19 and HDA6 but also provide insights into the unique feature of SIN3-type HDAC complexes in angiosperms.

Although the HDA19 histone deacetylase complex contains conserved accessory subunits, including SNLs, HDC1, and MSI1, the developmental phenotypes observed in the hda19 mutant are obviously different from those in the mutants of the conserved accessory subunits (Ning et al, 2019). In contrast, the developmental phenotypes in the hda19 mutant are highly similar to those observed in the hdip1/2/3 mutant. Moreover, the hdip1/2/3 and hda19 mutants also share the phenotypes in ABA sensitivity and drought tolerance. These phenotypic similarities suggest that the function of HDA19 is tightly associated with HDIP1/2/3 in Arabidopsis plants. This is further supported by their co-occupancy on chromatin and their co-regulation of histone deacetylation and transcriptional repression at the whole-genome level. Our genome-wide analysis indicates that HDIP1/2/3 and

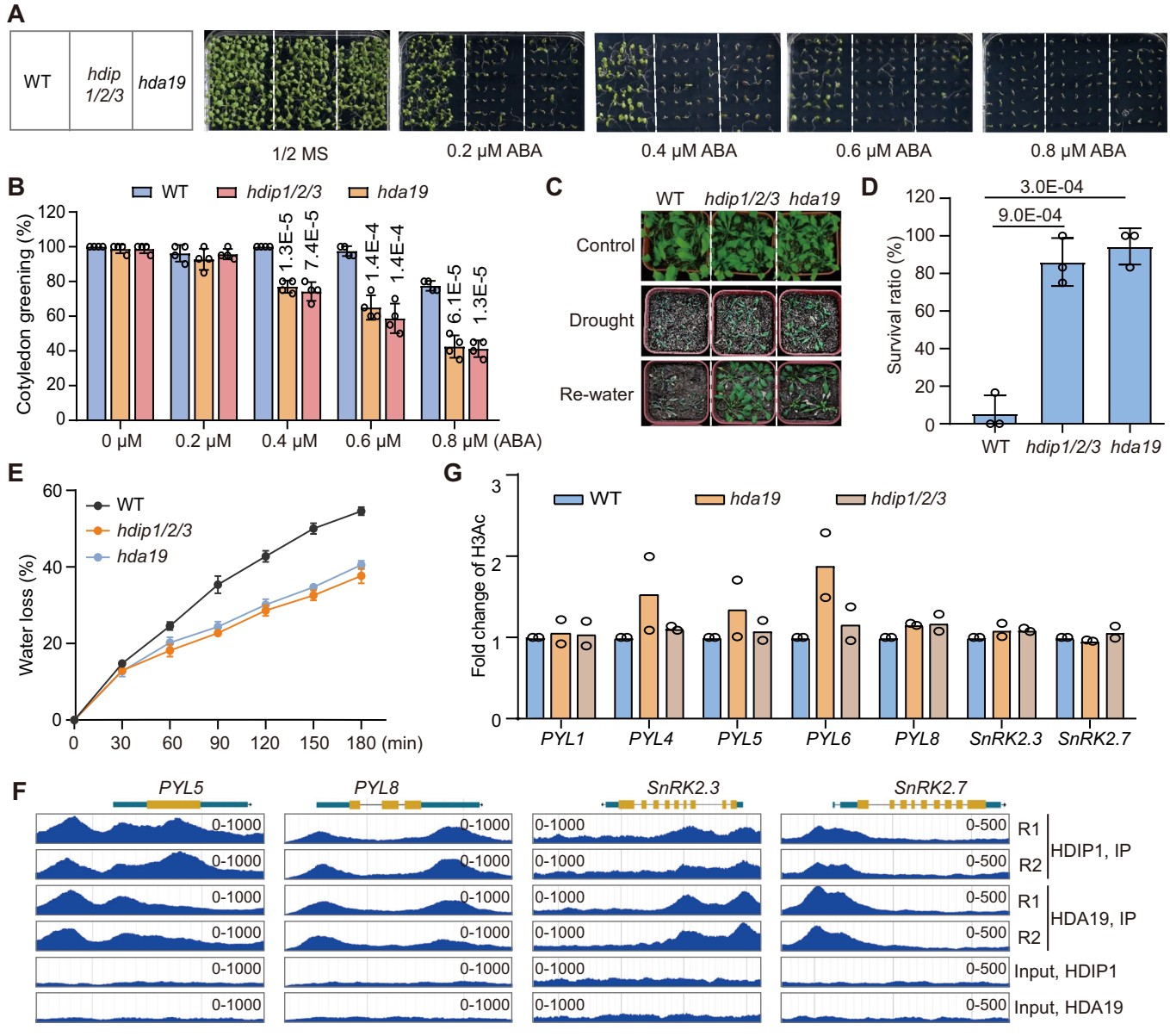

**Figure 7.  HDIP1/2/3 and HDA19 are involved in the regulation of ABA response and drought tolerance in Arabidopsis.**

(A) Germination phenotypes of 10-day-old wild-type, *hdip1/2/3*, and *hda19* mutant plants with or without ABA treatment. Seeds were cultivated on 1/2 MS medium with different concentrations of ABA. (B) Statistical analysis of cotyledon greening rate measured 12 days after the start of cultivation (*n* = 5). The percentage was obtained from 20 plants. Values are means ± SD. *P* values were determined by two-tailed Student's *t* test. (C) The drought tolerance phenotypes of wild-type, *hdip1/2/3*, and *hda19* mutant plants. Twelve-day-old plants were cultivated in water-saturated soil and were not provided with additional water until wild-type wilted. Subsequently, the plants were rewatered for 2 days. (D) Statistical analysis of the survival rate of the wild-type, *hdip1/2/3*, and *hda19* at 2 days after rehydration. Twelve plants were grouped for calculating the survival rate. Values are means ± SD of three biological replicates. *P* values were determined by two-tailed Student's *t* test. (E) Relative water loss of the wild-type, *hdip1/2/3*, and *hda19* mutants. The detached leaves were allowed to dry naturally under room temperature conditions and weighed at the indicated time points. Values are means ± SD of three biological replicates. (F) Genome browser view of HDIP1 and HDA19 ChIP-seq signals at representative ABA signaling pathway genes. Two independent biological replicates of the ChIP-seq results are indicated. (G) H3Ac ChIP-seq levels of the ABA signaling pathway genes in the wild-type, *hda19*, and *hdip1/2/3* mutants. Data are based on two independent biological replicates. Source data are available online for this figure.

HDA19 can mediate transcriptional repression via both histone deacetylation-dependent and -independent mechanisms. Previous studies have indicated that HDA19, HDC1, and MSI1 repress the expression level of specific ABA and stress-responsive genes exclusively by mediating histone deacetylation (Perrella et al, 2013; Mehdi et al, 2016). Our GO analysis of the RNA-seq data indicates that upregulated DEGs identified in the *hdip1/2/3* and *hda19* mutants are closely associated with the ABA signaling pathway. Since these ABA signaling pathway components can boost ABA response and drought tolerance (Yoshida et al, 2014), the increased expression of these components offers a plausible explanation for the increase of ABA sensitivity and drought

tolerance in the *hdip1/2/3* and *hda19* mutants. Interestingly, our whole-genome analysis indicated that the increased expression levels of numerous ABA and stress-responsive genes in the *hdip1/2/3* and *hda19* mutants are not accompanied by altered histone acetylation. This suggests that at the whole-genome level, the histone deacetylation-independent mechanism plays a crucial role for the SIN3-type histone deacetylase complexes in mediating transcriptional repression.

Given that HDIP1/2/3 orthologs are unique to angiosperms, they likely play a crucial role in the adaptation of these plants to adverse terrestrial conditions. Angiosperms, which constitute over 90% of all land plants, dominate most terrestrial ecosystems (Christenhusz and Byng, 2016). To cope with a variety of stress conditions, angiosperms have evolved a diverse array of stress-responsive genes, and these genes must be repressed under non-stress conditions to facilitate normal plant growth (Qin et al, 2011). The whole-genome analysis has revealed that histone acetylation is extensively accumulated at protein-coding genes, including many stress-responsive genes that are maintained at low expression levels in the absence of stress (Feng et al, 2021). Previous studies have indicated that high expression levels of stress-responsive genes can suppress plant growth and development (Li et al, 2022; Kishor et al, 2022; Wang et al, 2023a), which is consistent with our finding that the *hdip1/2/3* and *hda19* mutants exhibit severe developmental defects. Since these stress-responsive genes need to be rapidly induced following stress stimuli, the maintenance of high levels of histone acetylation may represent an adaptive mechanism that primes the chromatin for the subsequent activation of these genes. Therefore, we predict that the histone deacetylation-independent function of SIN3-type histone deacetylase complexes is essential for the chromatin to sustain the low transcriptional activity of stress-inducible genes under non-stress conditions while retaining the capacity for their timely induction. This contributes to the regulatory flexibility of stress-inducible genes, which is crucial for the survival of angiosperms under terrestrial conditions. Considering the pivotal role of HDIP1/2/3 within SIN3-type histone deacetylase complexes, we hypothesize that, as angiosperm-specific subunits, HDIP1/2/3 are critical for the SIN3-type histone deacetylase complexes to selectively target genes for transcriptional repression through histone deacetylation-dependent or -independent mechanisms. Further research is necessary to elucidate how HDIP1/2/3 cooperate with other subunits within the SIN3-type histone deacetylase complexes to distinguish between target genes that require deacetylation and those that do not.

Our study indicates that the HDIP1 protein binds to DNA in a non-sequence-specific manner, with a preference for four-way junction DNA that mimics the DNA entry and exit sites of nucleosomes (Varga-Weisz et al, 1993). Because the DNA-binding domain of HDIP1 is dispensable for the association of HDIP1 and HDA19 with chromatin, the binding of HDIP1 to nucleosomal DNA may facilitate HDA19-mediated histone deacetylation after the complex associates with chromatin. Moreover, yeast histone H1 and high mobility group (HMG) proteins, which are known for their involvement in chromatin compaction and transcriptional repression, also exhibit a preference for binding to four-way junctions and the DNA entry and exit sites of nucleosomes (Bianchi et al, 1989; Zhou et al, 2013a). The binding of HDIP1 to four-way junction DNA and nucleosomal DNA implies that HDIP1/2/3 could modify the conformation of HDA19 on the nucleosome, thereby allowing the catalytic site of HDA19 to effectively engage with acetylated histone tails for deacetylation. Moreover, since our study has also established that both HDA19 and HDIP1/2/3 can repress transcription through a mechanism that is independent of histone deacetylation, it is reasonable to infer that, similar to H1 and HMG proteins, HDIP1/2/3 may contribute to chromatin compaction by working in concert with other subunits of histone deacetylase complexes. Taken together, this study not only identifies previously unknown angiosperm-specific subunits of HDAC complexes but also provides insights into their roles in both histone deacetylation-dependent and -independent transcriptional repression.

# Methods

**Reagents and tools table**

| Reagent/resource | Reference or source | Identifier or catalog number |
|---|---|---|
| **Experimental models** | | |
| MCF7cell (Human): *Flag-ESR1* | Wang and Han, 2024 | N/A |
| Arabidopsis: *hdip1, hdip2, hdip3, hdip1/3, hdip1/2, hdip2/3, hdip1/2/3* | This study | N/A |
| Arabidopsis: *hda19* | Ning et al, 2019 | N/A |
| Arabidopsis: *1305-HDA19-Flag/ hda19* | Ning et al, 2019 | N/A |
| Arabidopsis: *1305-HDA6-Flag/ Col-0* | Ning et al, 2019 | N/A |
| Arabidopsis: *1305-MSI1-Flag/ Col-0* | Ning et al, 2019 | N/A |
| Arabidopsis: *1305-HDC1-Flag/ Col-0* | Ning et al, 2019 | N/A |
| Arabidopsis: *1305-SNL3-Myc/ Col-0* | Ning et al, 2019 | N/A |
| Arabidopsis: *1305-SNL4-Flag/ Col-0* | This study | N/A |
| Arabidopsis: *1305-SNL5-Flag/ Col-0* | This study | N/A |
| Arabidopsis: *1305-SNL6-Flag/ Col-0* | This study | N/A |
| Arabidopsis: *1305-HDIP1-Flag/ Col-0* | This study | N/A |
| Arabidopsis: *1305-HDIP1-Flag/ hdip1/2/3* | This study | N/A |
| Arabidopsis: *1305-HDIP2-Flag/ Col-0* | This study | N/A |
| Arabidopsis: *1305-HDIP3-Flag/ Col-0* | This study | N/A |
| Arabidopsis: *1305-HDA19-Flag/ hda19hdip1/2/3* | This study | N/A |
| Arabidopsis: *1305-HDIP1-Δa2/3-Flag/Col-0* | This study | N/A |
| Arabidopsis: *1305-HDIP1-ΔC-Flag/Col-0* | This study | N/A |
| Arabidopsis: *1305-HDIP1-ΔDBD-Flag/Col-0* | This study | N/A |

| Reagent/resource | Reference or source | Identifier or catalog number |
|---|---|---|
| Arabidopsis: *1305-HDIP1-ΔDBD-Flag/hdip1/2/3* | This study | N/A |
| Escherichia coli: Mach1-T1 | Biomed | Cat#BC105-02 |
| Escherichia coli: T7pLysY | Biomed | Cat#BC207-02 |
| Agrobacterium: GV3101 | Zoman | Cat#ZC1407 |
| **Recombinant DNA** | | |
| pCAMBIA1305-SNL4-Flag | This study | N/A |
| pCAMBIA1305-SNL5-Flag | This study | N/A |
| pCAMBIA1305-SNL6-Flag | This study | N/A |
| pCAMBIA1305-HDIP1-Flag | This study | N/A |
| pCAMBIA1305-HDIP1-Flag | This study | N/A |
| pCAMBIA1305-HDIP2-Flag | This study | N/A |
| pCAMBIA1305-HDIP3-Flag | This study | N/A |
| pCAMBIA1305-HDA19-Flag | This study | N/A |
| pCAMBIA1305-HDIP1-Δα2/3-Flag (631–720 aa) | This study | N/A |
| pCAMBIA1305-HDIP1-ΔC-Flag (631–896 aa) | This study | N/A |
| pCAMBIA1305-HDIP1-ΔDBD-Flag (631–896 aa) | This study | N/A |
| pCAMBIA1305-HDIP1-ΔDBD-Flag (766–826 aa) | This study | N/A |
| PGADT7/PGBKT7-HDIP1 | This study | N/A |
| PGADT7/PGBKT7-HDIP3 | This study | N/A |
| PGADT7/PGBKT7-HDIP1-1 (1–448 aa) | This study | N/A |
| PGADT7/PGBKT7-HDIP1-2 (449–896 aa) | This study | N/A |
| PGADT7/PGBKT7-HDIP1-3 (449–630 aa) | This study | N/A |
| PGADT7/PGBKT7-HDIP1-4 (631–720 aa) | This study | N/A |
| PGADT7/PGBKT7-HDIP1-5 (721–896 aa) | This study | N/A |
| PGADT7/PGBKT7-SNL5 | This study | N/A |
| PGADT7/PGBKT7-SNL5-1 (1–725 aa) | This study | N/A |
| PGADT7/PGBKT7-SNL5-2 (726–1167 aa) | This study | N/A |
| PGADT7/PGBKT7-SNL5-3 (1–191 aa) | This study | N/A |
| PGADT7/PGBKT7-SNL5-4 (192–305 aa) | This study | N/A |
| PGADT7/PGBKT7-SNL6 | This study | N/A |
| PGADT7/PGBKT7-SNL6-1 (1–680 aa) | This study | N/A |
| PGADT7/PGBKT7-SNL6-2 (681–1122 aa) | This study | N/A |
| PGADT7/PGBKT7-SNL6-3 (1–179 aa) | This study | N/A |
| PGADT7/PGBKT7-SNL6-4 (180–282 aa) | This study | N/A |
| PGEX6P-1-GST-HDIP1-3 (449–630 aa) | This study | N/A |
| PGEX6P-1-GST-HDIP1-4 (631–720 aa) | This study | N/A |
| PGEX6P-1-GST-HDIP1-9 (1–300 aa) | This study | N/A |
| PGEX6P-1-GST-HDA19 | This study | N/A |
| PGEX6P-1-GST-HDA19-1 (1–400 aa) | This study | N/A |
| PGEX6P-1-GST-HDA19-2 (400–501 aa) | This study | N/A |
| PET30a-MBP-HDIP1-6 (721–766 aa) | This study | N/A |
| PET30a-MBP-HDIP1-7 (766–827 aa) | This study | N/A |
| PET30a-MBP-HDIP1-8 (827–896 aa) | This study | N/A |
| PET30a-MBP-HDIP3-DBD (806–930 aa) | This study | N/A |
| PET30A-MBP-HDIP3-C (906–1005 aa) | This study | N/A |
| PET28a-His-SNL5-3 (1–191 aa) | This study | N/A |
| PET28a-His-SNL6-3 (1–179 aa) | This study | N/A |
| PET28a-His-HDC1 | This study | N/A |
| PET28a-His-MSI1 | This study | N/A |
| GAL4-BD-HDIP1 | This study | N/A |
| GAL4-BD-HDIP2 | This study | N/A |
| GAL4-BD-HDA19 | This study | N/A |
| GAL4-HDIP1 | This study | N/A |
| GAL4-BD-HDIP1-1 (1–448 aa) | This study | N/A |
| GAL4-BD-HDIP1-2 (449–630 aa) | This study | N/A |
| GAL4-BD-HDIP1-3 (631–720 aa) | This study | N/A |
| GAL4-BD-HDIP1-4 (721–896 aa) | This study | N/A |
| **Antibodies** | | |
| Anti-Flag | Sigma | Cat#F1804 |
| Anti-H3Ac | Millipore | Cat#06-599 |
| Anti-H3K9Ac | Millipore | Cat#07-352 |
| Anti-H3K14Ac | Millipore | Cat#07-353 |
| Anti-H4k5Ac | Abcam | Cat#ab51997 |
| Anti-GST | Abcam | Cat#ab19256 |

| Reagent/resource | Reference or source | Identifier or catalog number |
|---|---|---|
| Anti-MBP | Abcam | Cat# ab9084 |
| Anti-His | TransGen | Cat# HT501-01 |
| Protein G Dynabeads | Invitrogen | Cat#10004D |
| Protein A Dynabeads | Invitrogen | Cat#10001D |
| Flag M2 affinity agarose gel | Sigma | Cat#A2220 |
| **Oligonucleotides and other sequence-based reagents** | | |
| PCR primers | This study | Dataset EV6 |
| ChIP-qPCR primers | This study | Dataset EV6 |
| EMSA probes | This study | Dataset EV6 |
| **Chemicals, enzymes, and other reagents** | | |
| Nextflex™ Rapid DNA-seq Kit | Bioo Scientific | Cat#5144-08 |
| ProteoSliver Silver Stain Kit | Sigma | Cat#PROT-SIL1 |
| 5×All-In-One RT Master Mix | Abm | Cat#G492 |
| One-Step Cloning Kit | Vazyme Biotech | Cat#C112 |
| **Software** | | |
| DNAMAN 6.0 | https://www.lynnon.com/ | N/A |
| MEGA 7.0 | https://www.megasoftware.net/ | N/A |
| NLStradamus | http://www.moseslab.csb.utoronto.ca/NLStradamus/ | N/A |
| Alphafold2 | https://alphafold.ebi.ac.uk/ | N/A |
| Venn diagram | Jia et al, 2021 | N/A |
| GraphPad prism (v8) | https://www.graphpad.com/ | N/A |
| DAVID website | https://davidbioinformatics.nih.gov/home.jsp | N/A |
| DeepTools (v3.5.1) | https://deeptools.readthedocs.io/en/develop/content/list_of_tools.html | N/A |
| HISAT2 (v2.2.0) | Kim et al, 2019 | N/A |
| FeatureCounts (v2.0.2) | Liao et al, 2014 | N/A |
| R package | Robinson et al, 2010 | N/A |
| Bowtie2 | Langmead and Salzberg, 2012 | N/A |
| MACS2 | Zhang et al, 2008 | N/A |
| SICER2 | Zang et al, 2009 | N/A |
| **Other** | | |
| Illumina Novaseq 6000 | Illumina | N/A |

## Plant materials and plasmid constructs

The Arabidopsis plants utilized in this study are all from the wild-type Col-0 ecotype. The *hda19* mutant (SALK_139445) was previously reported (Feng et al, 2021). The *hdip1/3* and *hdip2* mutants were acquired through the CRISPR-Cas9 genome editing system (Wang et al, 2015). The *hdip1/2/3* triple mutants were generated by introducing the *hdip2* mutation into the *hdip1/3* double mutant. The single mutants *hdip1* and *hdip3* and the double mutants *hdip1/2* and *hdip2/3* were generated by crossing. The full-length genomic sequences of *SNL4*, *SNL5*, *SNL6*, *HDIP1*, *HDIP2*, and *HDIP3*, as well as the genomic sequences encoding truncated HDIP1 proteins including HDIP1-Δα2/3 (631–720 aa), HDIP1-ΔC (631–896 aa), HDIP1-ΔDBD (766–826 aa), along with their native promoters were amplified and inserted into the *pCAMBIA1305* vector, which includes a C-terminal Flag tag. The resulting constructs were then transformed into both wild-type and mutant plants using the flower-dipping method. Transgenic plants were selected on the MS (Murashige and Skoog) medium supplemented with hygromycin (30 mg/L) and ampicillin (50 mg/L). *MSI1-Flag*, *HDC1-Flag*, *SNL3-Myc*, *HDA19-Flag*, *HDA6-Flag* transgenic plants have been previously described (Ning et al, 2019). The primers used for the construction of these vectors are listed in Dataset EV5. All plants were grown on MS or 1/2 MS medium plates under long-day conditions (16 h light/8 h darkness) at 22 °C.

## Yeast two-hybrid assays

For Y2H assays, full-length and truncated coding sequences of *HDIP1*, *HDIP3*, *HDA19*, *SNL5*, *SNL6*, *HDC1*, and *MSI1* were amplified through PCR and subsequently cloned into *pGADT7* or *pGBKT7* vectors using the One-Step Cloning Kit (Vazyme Biotech, C112). The primers used for the construction of these vectors are listed in the Dataset EV5. The *pGADT7* and *pGBKT7* vectors were transformed into the AH109 (mating type a) and Y187 (mating type α) yeast strains, respectively. The AH109 strain was grown on synthetic dropout medium lacking Leu (SD-L), and the Y187 strain was grown on synthetic dropout medium lacking Trp (SD-W). Positive colonies on SD-L and SD-W were co-cultured in liquid YPDA medium for 16 h at 22 °C to facilitate mating. The resulting cells were collected and then plated on SD medium lacking Trp and Leu (SD-WL). After a 2-day incubation period, the positive colonies on SD-WL were resuspended in sterile distilled water and spotted on SD medium lacking Trp, Leu, and His (SD-WLH), supplemented with 3-amino-1,2,4-triazole (3-AT).

## Protein purification and pull-down assay

The full-length and truncated coding sequences of *HDIP1*, *HDA19*, *HDA6*, *SNL5*, *SNL6*, *HDC1*, and *MSI1* were individually cloned into pET30a, pGEX6P-1 or pET28a vectors. The primers used for the construction of these constructs are listed in Dataset EV5. The recombinant fusion proteins were expressed in the *Escherichia coli* strain BL21 (DE3) and induced with 0.1 M isopropyl-b-D-thiogalactopyranoside (IPTG) at 16 °C for 16 h. The cells were collected and resuspended in either GST-lysis buffer (20 mM Tris-HCl pH 8.0, 150 mM NaCl, and 1 mM DTT), His-lysis buffer (20 mM Tris-HCl pH 8.0, 500 mM NaCl, 20 mM imidazole, and 1 mM DTT), or MBP-lysis buffer (20 mM Tris-HCl pH 8.0, 200 mM NaCl, 1 mM EDTA, and 1 mM DTT). After sonication and centrifugation at 14,000 × *g* for 1 h at 4 °C, the supernatant was incubated with GST beads, MBP beads, or His beads for 2 h at 4 °C. The bound proteins were washed four times with lysis buffer. The proteins tagged with GST, His, and MBP were eluted using GST

elution buffer (GST-lysis buffer containing 20 mM glutathione), His elution buffer (His-lysis buffer containing 250 mM imidazole), and MBP elution buffer (MBP-lysis buffer containing 10 mM maltose), respectively. The purified proteins were resolved on SDS-PAGE gel and visualized by staining with Coomassie brilliant blue.

The in vitro pull-down assay was performed as previously described (Du et al, 2023). Pairs of proteins, each tagged with either GST, MBP, or His, were combined and incubated with either GST or MBP beads for 2 h at 4 °C. Subsequently, the beads were washed five times with their respective lysis buffers to remove unbound proteins. The bound proteins were then eluted using the appropriate elution buffers specific for GST or MBP. The eluted proteins were separated on SDS-PAGE gel, transferred onto a membrane for western blot analysis, and detected using anti-GST (Abcam, ab19256, dilution-1:5000), anti-MBP (Abcam, ab9084, dilution-1:5000), or anti-His (TransGen, HT501-01, dilution-1:5000).

## RNA-seq and quantitative RT-PCR

Total RNA was extracted from 20-day-old wild-type and mutant plants using TRIzol reagent (Invitrogen, 15596018), and then sent to Novogene for library construction and sequencing (Illumina NovaSeq 6000, PE150). The RNA-seq data were obtained from two independent biological replicates. The processing of the raw sequencing reads involved the removal of adapter sequences and the filtering out of low-quality reads. The clean reads were then mapped to the Arabidopsis genome (TAIR10) using the software HISAT2 (v2.2.0) (Kim et al, 2019). Uniquely mapped reads were applied for expression analysis using the featureCounts algorithm (v2.0.2) (Liao et al, 2014). Differentially expressed genes (DEGs) were identified using cuffdiff (P value < 0.05, $\log_2$FC (fold change) >1 or < −1) with the R package edgeR (v3.32.1) (Robinson et al, 2010). Venn diagrams were generated using a Venn diagram tool (Jia et al, 2021). The heatmap of DEGs was drawn using R package gplots (v3.1.1). GO enrichment analysis was conducted using the online DAVID tool (https://david.ncifcrf.gov/tools.jsp) (Sherman et al, 2022). For quantitative RT-PCR, 2 µg RNA was subjected to cDNA synthesis using 5×All-In-One RT Master Mix (Abm, G492). The primers used for RT-PCR are listed in Dataset EV5.

## ChIP-seq and quantitative ChIP-PCR

Chromatin immunoprecipitations (ChIP) assay was performed using 12-day seedlings of HDIP1-Flag and HDA19-Flag transgenic plants according to the previous method with minor modifications (Guo et al, 2022). The ChIP-seq for H3Ac was performed using 20-day-old WT and mutant seedlings. In brief, 4 g samples were cross-linked with 1% formaldehyde (Sigma-Aldrich, F8755) for 12 min and the reaction was flash stopped with 0.125 M glycine under vacuum. After washing the cross-linked materials with ice-cold ddH$_2$O, the samples were ground in liquid nitrogen and resuspended in 35 ml ice-cold lysis buffer (0.4 M sucrose, 10 mM Tris-HCl pH 8.0, 10 mM MgCl$_2$, 0.25% Triton X-100, 1 mM DTT, 0.1 mM PMSF, and 1× protease inhibitor cocktail, Roche) for 30 min at 4 °C. Resuspended tissue was filtered twice through Microcloth (Millipore, 475855) and centrifuged for 20 min at 1500 × g at 4 °C.

The extracted nuclei were washed three times with wash buffer (10 mM Tris-HCl pH 8.0, 10 mM MgCl$_2$, 0.25 M sucrose, and 1% Triton X-100). The pellet was then resuspended in buffer (10 mM

Tris-HCl pH 8.0, 10 mM MgCl$_2$, 1.7 M sucrose, 0.25% Triton X-100, 1 mM DTT, 0.1 mM PMSF, and 1× protease inhibitor cocktail, Roche) and centrifuged for 1 h at 14,000 × g at 4 °C. Subsequently, the nuclei pellet was suspended in sonication buffer (50 mM Tris-HCl pH 8.0, 10 mM EDTA, 1% SDS, 1 mM DTT, 0.1 mM PMSF, and 1× protease inhibitor cocktail, Roche) and sonicated using the Bioruptor (Diagenode). After centrifugation, the supernatant was diluted with dilution buffer (20 mM Tris-HCl pH 8.0, 2 mM EDTA, 200 mM NaCl, 1 mM DTT, 0.1 mM PMSF, and 1× protease inhibitor cocktail, Roche) and incubated with either Flag antibody (Sigma, F1804) or H3Ac antibody (Millipore, 06-599) overnight. Then 40 µl/sample of Protein G Dynabeads (Invitrogen, 10004D) or Protein A Dynabeads (Invitrogen, 10001D) were added and incubated for 2 h at 4 °C. The beads were washed sequentially with low-salt buffer (20 mM Tris-HCl pH 8.0, 150 mM NaCl, 0.1% SDS, 1% Triton X-100, and 2 mM EDTA), high-salt buffer (20 mM Tris-HCl pH 8.0, 500 mM NaCl, 0.1% SDS, 1% Triton X-100, and 2 mM EDTA), LiCl buffer (0.25 M LiCl, 1% sodium deoxycholate, 1% NP-40, 10 mM Tris-HCl pH 8.0, and 1 mM EDTA), and TE buffer (10 mM Tris-HCl pH 8.0 and 1 mM EDTA) for 10 min at 4 °C with rotation. The protein-DNA complex was eluted with elution buffer (1% SDS and 0.1 mM NaHCO$_3$) at 65 °C and reverse cross-linked overnight. The DNA was extracted using the phenol/chloroform/isoamyl alcohol (25:24:1) mixture.

The purified DNA was utilized to construct sequencing libraries using the Nextflex™ Rapid DNA-seq Kit (New England Biolabs, USA), and the libraries were sent to Novogene for sequencing on the Illumina NovaSeq 6000 platform (PE150). The clean reads were obtained by filtering out adapter sequences and low-quality reads. These clean reads were mapped to the TAIR10 Arabidopsis genome using Bowtie2 (v2.3.4), allowing up to one mismatch (Langmead and Salzberg, 2012). Duplicated reads were removed using Picard Tool (v2.23.0) with MarkDuplicates option. Enriched peaks were identified by comparing then to the input peaks using MACS2 (v2.2.7.1) (Zhang et al, 2008). For H3Ac ChIP-seq, differentially enriched peaks between the wild-type and mutants were identified by SICER2 (v1.0.2) using cuffdiff (FDR < 0.05, FC < 0.8 or >1.2) (Zang et al, 2009). Read counts were normalized to reads per kilobase per million mapped reads (RPKM) based on the number of clean reads mapped to the genome in each library. Only the reads that uniquely mapped to the genome were retained for further analysis. The experiment was conducted with two independent biological replicates, with the input file serving as a control. Heatmaps were generated using DeepTools (v3.5.1). Box plots, scatter plots, and volcano plots were drawn with the R package ggplot2. Venn diagrams were produced using a Venn diagram tool (Jia et al, 2021). GO analysis was performed using the online DAVID website.

For quantitative ChIP-PCR, MCF7 cells (human) with Flag-ESR1 (Estrogen Receptor 1) were used as a spike-in control to normalize the HDA19-Flag enrichment level in hda19 and hda19hdip1/2/3 mutants. After sonication, 25% of spike-in chromatin from MCF7 cells was mixed with the target chromatin, and the mixture was incubated with Flag antibody (Sigma, F1804) coupled with Dynabeads Protein G (Invitrogen, 10004D) at 4 °C overnight for immunoprecipitation. GREB1 represents the positive locus in the spike-in genome (Wang and Han, 2024). The primers used for ChIP-PCR are listed in Dataset EV5.

Quantitative ChIP-PCR in Arabidopsis mesophyll protoplast was performed as previously described with minor modifications (Lee et al, 2017). Briefly, the isolation of Arabidopsis mesophyll protoplast and

DNA transfection were followed by a previously described method (Yoo et al, 2007). The transfected protoplast was centrifuged at $100 \times g$ for 2 min at room temperature, followed by removal of the supernatant. The pellet was washed twice with 1 ml of 1× PBS buffer (pH 7.4). Then 27 μl of 37% formaldehyde was added to 1 ml of 1× PBS buffer (pH 7.4) to crosslink the sample on a rotor for 10 min at room temperature. The reaction was immediately stopped with 0.1 M glycine on a rotor for 5 min at room temperature. After centrifugation at $1500 \times g$ for 5 min at 4 °C, the pellet was rinsed twice with 1 ml of ice-cold 1× PBS buffer (pH 7.4). Removing the supernatant, the pellet was resuspended in 1 ml harvest buffer (100 mM Tris-HCl pH 9.4, 10 mM DTT) and incubated at 30 °C for 15 min. The sample was centrifuged and resuspended in 1 ml of ice-cold 1× PBS buffer (pH 7.4). The nuclei were washed with wash buffer (10 mM HEPES pH 6.5, 10 mM EDTA, 0.5 mM EGTA, and 0.25% Triton X-100) three times and resuspended in ice-cold nuclei lysis buffer (50 mM Tris-HCl pH 8.0, 10 mM EDTA, 1% SDS and 1× protease inhibitor cocktail), followed by sonication for six cycles (10 s ON and 30 s OFF) using the Bioruptor (Diagenode). The sonicated chromatin was diluted tenfold with dilution buffer and incubated with H3Ac antibody (Millipore, 06-599) and Dynabeads Protein A (Invitrogen, 10001D). The DNA was extracted using phenol/chloroform/isoamyl alcohol (25:24:1). The primers used for ChIP-PCR are listed in Dataset EV5.

## AP-MS analysis

Five grams of a mixture of 12-day-old seedlings and adult plant inflorescences was ground into powder in liquid nitrogen and then resuspended in 20 mL ice-cold lysis buffer (50 mM Tris-HCl pH 7.5, 150 mM NaCl, 5 mM MgCl$_2$, 10% glycerol, 0.1% NP-40, 1 mM DTT, 1 mM PMSF, and 1× protease inhibitor cocktail, Roche) for 20 min at 4 °C with rotation. The suspended tissues were centrifuged for 15 min at $14,000 \times g$ at 4 °C and filtered once through two layers of Miracloth to obtain the supernatant. The supernatant was incubated with 100 μL anti-Flag M2 affinity agarose gel (Sigma, A2220) at 4 °C for 2.5 h. The agarose beads were washed five times with lysis buffer, and the bound proteins were eluted with 3×Flag peptide (Sigma, F4799). The eluted proteins were run on a 10% SDS-PAGE gel and stained using the ProteoSliver silver stain kit following the manufacturer's instructions (Sigma, PROT-SIL1). The stained proteins were then excised from the gel and analyzed by mass spectrometry as previously described (Zhang et al, 2012). The heatmap of AP-MS data was plotted using GraphPad prism (v8).

## HDAC assay

The histone deacetylation (HDAC) assay in vitro was performed as previously described (Guan et al, 2023). A quantity of 0.5 g of HDA19-containing complexes were purified from *HDA19-Flag* transgenic plants using anti-Flag M2 affinity agarose gel (Sigma, A2220). The same purification procedure was applied to wild-type Col-0, and the resulting proteins served as a negative control. The HDA19-containing complexes, along with the negative control, were incubated with 3 μg of calf thymus histones (Roche, 10223565001) at 30 °C for 1 h in a histone deacetylation reaction buffer (20 mM HEPES pH 7.5, 150 mM NaCl, 0.2 mg/ml BSA, 1 mM DTT and 1× protease inhibitor cocktail, Roche). The reaction was stopped with 80 mM EDTA and 5×SDS loading buffer at 98 °C for 10 min. The samples were run on 10–12% SDS-PAGE gel and

detected using antibodies against H3Ac (Millipore, 06-599), H3K9Ac (Millipore, 07-352), H3K14Ac (Millipore, 07-353) and H4K5Ac (Abcam, ab51997).

## EMSA

Four-way junction DNA was constructed using four single-stranded oligonucleotides according to a previously described method (Elborough and West, 1990). The oligonucleotides were synthesized and fluorescently labeled with Cy5 dye. Four purified strands were annealed by incubation at 95 °C for 5 min, followed by gradual cooling. The junction DNA was run on agarose gels for separation and purification. The linear double-stranded DNA was created by annealing oligonucleotides that were labeled with Cy5 dye. Nucleosome assembly was performed as previously described (Zheng et al, 2023). The oligonucleotides utilized in this study are listed in Dataset EV5.

The EMSA was conducted following a previously described method with minor modifications (Du et al, 2023). The binding reaction mixture, comprising purified proteins and either double-stranded DNA, four-way junction DNA, or nucleosomes, was incubated in binding buffer (10 mM Tris-HCl pH 7.5, 50 mM NaCl, 1 mM EDTA, 5% glycerol, 1 mg/mL bovine serum albumin, and 1 mM DTT) for 30 min at room temperature to allow for the complex formation. After the incubation, the samples were run on 6% native polyacrylamide gel containing 0.5×Tris-borate-EDTA buffer. The gels were then visualized using a Bio-Rad scanner according to the manufacturer's instructions.

## Nucleosomes assembly

The reconstitution of nucleosome was performed using the protocol previously described (Zheng et al, 2023). In brief, the full-length histones H2A, H2B, H3 and H4 were purified in *Escherichia coli* strain BL21. Then mixture of equal amount of H2A, H2B, H3 and H4 were dialyzed with refolding buffer (10 mM Tris-HCl pH 7.5, 2 M NaCl, 1 mM EDTA, and 1 mM DTT) at 4 °C overnight. The assembled octamer was concentrated into a volume of 500 μl and loaded onto AKTA system with HiLoad 16/600 Superdex-200 pg (289893-35; GE Healthcare) column to remove dimer and super polymers. The wisdom 601 plasmid (Ge et al, 2023) was used as a DNA template. The purified DNA and the octamer were mixed in 2 M NaCl and incubated on ice for 30 min. Then the mixture was dialyzed by gradient descent of salty ions from high-salt buffer (10 mM Tris-HCl pH 7.5, 2 M NaCl) to low-salt buffer (10 mM Tris-HCl pH 7.5, 0.2 M NaCl) for 36 h. Finally, the assembled nucleosome was dialyzed against TE buffer (10 mM Tris-HCl pH 8.0 and 1 mM EDTA).

## Dual-luciferase reporter assay

To generate the effector constructs, the full-length coding sequences of *HDIP1*, *HDIP2*, and *HDA19* were cloned into the *GAL4-BD* vector, either with or without the coding sequence of the transcriptional activator VP16. Arabidopsis mesophyll protoplasts were isolated from the leaves of the wild-type Col-0 ecotype, following a previously reported method (Yoo et al, 2007). The plasmids of the effector construct and the LUC reporter construct were delivered into the protoplasts through PEG–calcium-mediated transfection. The quantitative measurement of luminescence were

performed using the dual-luciferase reporter assay system as described by the manufacturer (Promega, E1910).

## ABA treatment and drought tolerance assays

For the ABA sensitivity assay, seeds of wild-type and mutants were cultivated on 1/2 MS medium supplemented with varying concentrations of ABA (0.2 μM, 0.4 μM, 0.6 μM, and 0.8 μM). The appearance of green cotyledons was determined 12 days after the start of cultivation. Data was from three independent biological experiments. In the drought tolerance assay, 12-day-old seedlings that had been cultivated on MS medium were transferred to soil that was saturated with water, and then subjected to drought treatment by withholding water while being grown under long-day conditions (16 h light/8 h darkness) at 22 °C. The drought-treated plants were rewatered after about 20 days without water. The survival rate was measured 2 days after rewatering. For the water-loss assay, rosette leaves were detached from 25-day-old plants, and their fresh weight was immediately measured at various time points post-detachment. The water-loss rates were calculated based on the reduction in fresh weight over the specified time intervals, following a previously described method (Verslues et al, 2006). The experiments were conducted with a minimum of three independent biological replicates.

## Protein structure, sequence alignment, and phylogenetic analysis

The amino acid sequences of proteins were obtained from National Center for Biotechnology Information (NCBI) and TAIR (https://www.arabidopsis.org/). The structure of proteins was predicted using Alphafold (https://alphafold.ebi.ac.uk/). NLS sequence were identified by NLStradamus (http://www.moseslab.csb.utoronto.ca/NLStradamus/). Multiple sequences alignments were carried out using DNAMAN (version 6) software. The phylogenetic tree was performed by MEGA (version 7) using the neighbor-joining methods with bootstrap (1000 replicates).

## Accession numbers

The accession number of genes examined in this study are accessible in the TAIR10 database and are listed as follows: AT1G69360 (*HDIP1*), AT1G26620 (*HDIP2*) and AT1G13940 (*HDIP3*), AT4G38130 (*HDA19*), AT5G63110 (*HDA6*), AT5G08450 (*HDC1*), AT5G58230 (*MSI1*), AT3G01320 (*SNL1*), AT5G15020 (*SNL2*), AT1G24190 (*SNL3*), AT1G70060 (*SNL4*), AT1G59890 (*SNL5*), AT1G10450 (*SNL6*), AT5G05440 (*PYL5*), AT5G53160 (*PYL8*), AT4G40010 (*SnRK2.7*), and AT5G66880 (*SnRK2.3*).

## Data availability

Raw RNA-seq and ChIP-seq data generated in this study have been deposited in the Gene Expression Omnibus (GEO) database with accession code GSE275989.

The source data of this paper are collected in the following database record: biostudies:S-SCDT-10_1038-S44318-025-00445-w.

## Peer review information

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

## Acknowledgements

The authors thank Ting Han and Li-Zhen Wang for providing human MCF7 cells. This work was supported by the National Natural Science Foundation of China (grant numbers: 32025003 and 32470628).

## Author contributions

**Na Liu**: Conceptualization; Data curation; Formal analysis; Validation; Investigation; Visualization; Methodology; Writing—original draft. **Jia-Xin Li**: Formal analysis; Validation; Investigation; Methodology. **Dan-Yang Yuan**: Data curation; Formal analysis. **Yin-Na Su**: Data curation; Formal analysis. **Pei Zhang**: Investigation; Methodology. **Qi Wang**: Investigation; Methodology. **Xiao-Min Su**: Investigation. **Lin Li**: Investigation; Methodology. **Haitao Li**: Investigation; Methodology. **She Chen**: Investigation; Methodology. **Xin-Jian He**: Conceptualization; Data curation; Supervision; Funding acquisition; Validation; Investigation; Writing—original draft; Project administration; Writing—review and editing.

Source data underlying figure panels in this paper may have individual authorship assigned. Where available, figure panel/source data authorship is listed in the following database record: biostudies:S-SCDT-10_1038-S44318-025-00445-w.

## Disclosure and competing interests statement

The authors declare no competing interests.

# Expanded View Figures

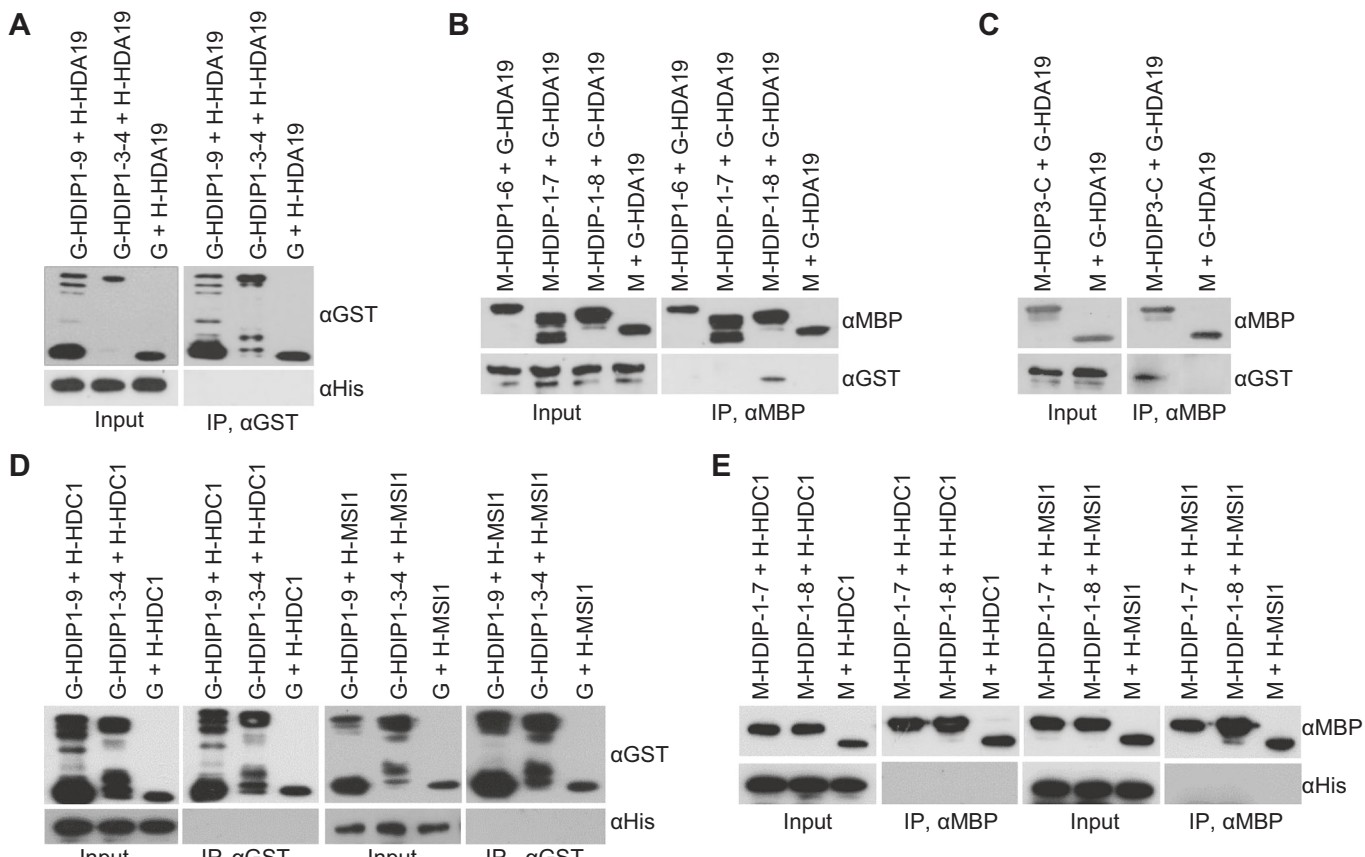

**Figure EV1. Interactions of truncated forms of HDIP1/3 with subunits of HDA19-containing histone deacetylase complexes as detected by pull-down assays.**

(A, B) The interaction between HDA19 and truncated forms of HDIP1 was detected by pull-down assays. (C) The interaction between HDA19 and HDIP3-C (906–1005 aa) was detected by pull-down assay. (D, E) The interaction between the truncated forms of HDIP1 and either HDC1 or MSI1 was detected by pull-down assays. G, GST; H, His; M, MBP. The truncated forms of HDIP1 are shown in Fig. 1C. Source data are available online for this figure.

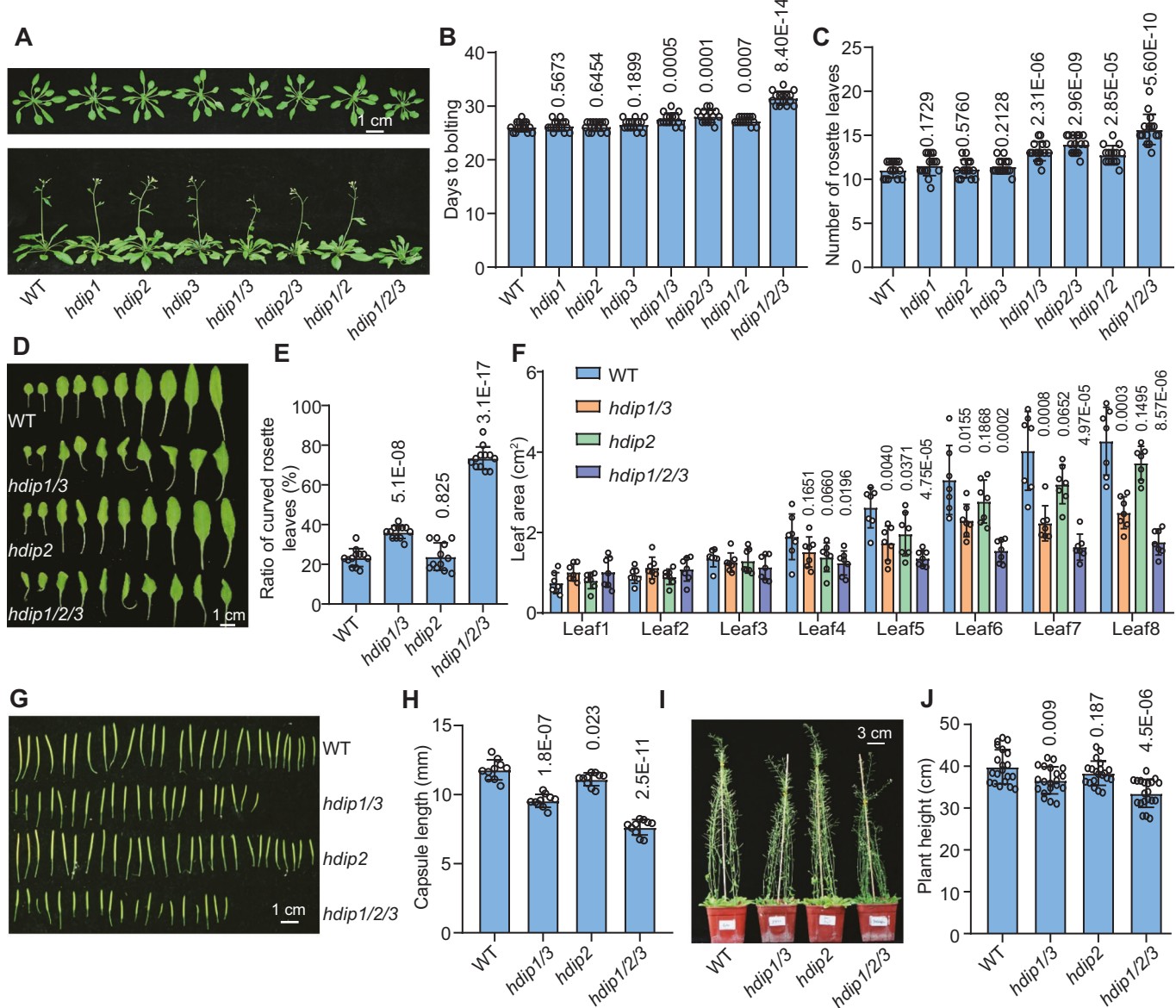

**Figure EV2.  HDIP1, HDIP2, and HDIP3 redundantly function in the regulation of plant growth and development.**

(A) Morphological phenotype of flowering time in the wild-type and mutants. (B, C) Statistical analyses of days to bolting ($n = 18$) (B), and the number of rosette leaves ($n = 18$) (C). Data are means ± SD. *P* values were determined by two-tailed Student's *t* test. (D) Morphological phenotype of rosette leaves from 32-day-old plants. (E, F) Statistical analysis of the ratio of curved rosette leaves ($n = 12$) (E), and leaf area ($n = 8$) (F). Data are means ± SD. *P* values were determined by two-tailed Student's *t* test. (G) Morphology of siliques in the wild-type and mutants. (H) Statistical analysis of the silique length. Data are means ± SD ($n = 10$). *P* values were determined by two-tailed Student's *t* test. (I) The plant height phenotype of the wild-type and mutants. (J) Statistical analysis of plant height. Data are means ± SD ($n = 20$). *P* values were determined by two-tailed Student's *t* test. Source data are available online for this figure.

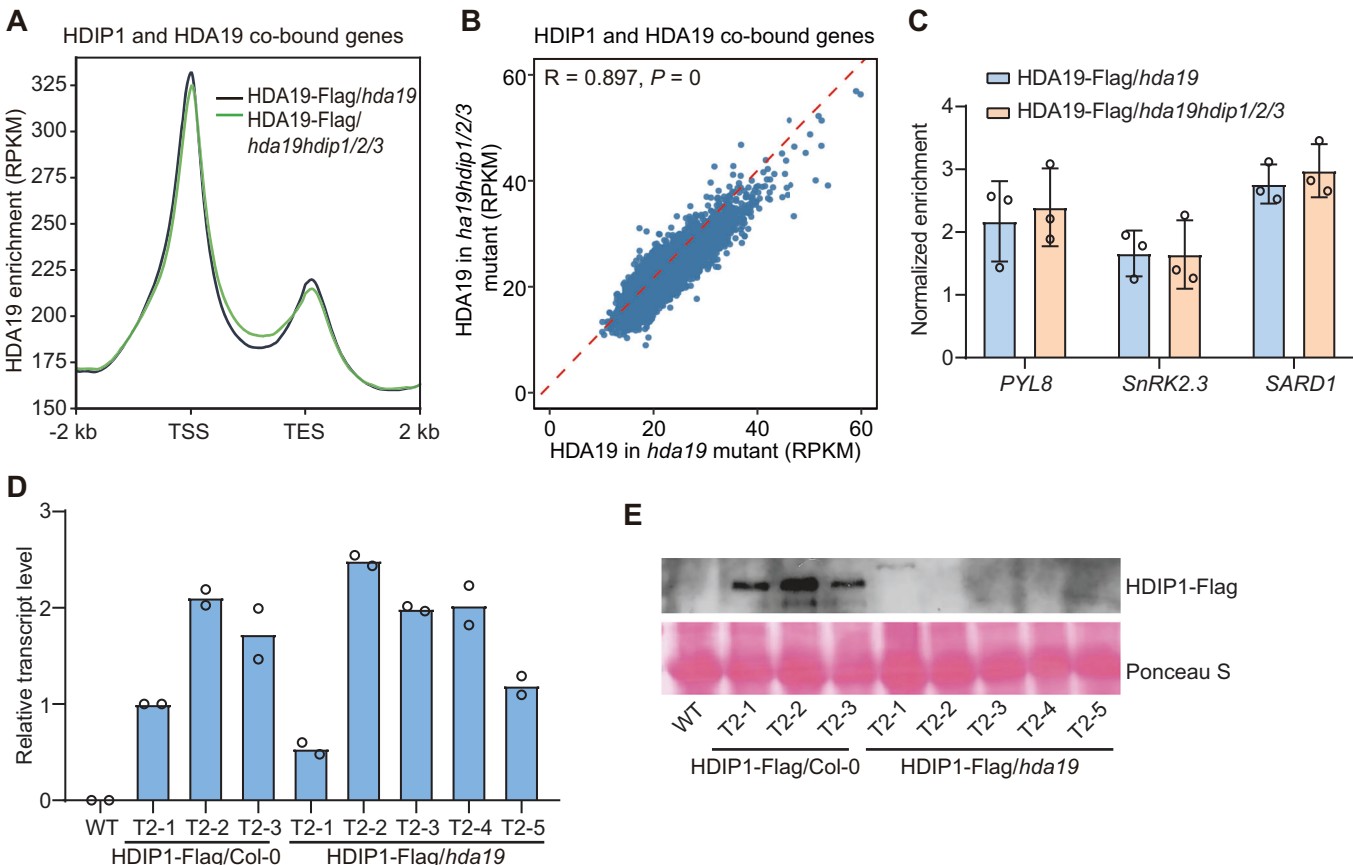

**Figure EV3. Determination of the effect of *hdip1/2/3* on the binding of HDA19 to chromatin.**

(**A**) Metaplots showing the ChIP-seq signals of HDA19-Flag in *hda19* and *hda19hdip1/2/3* mutant backgrounds at the target genes shared by HDA19 and HDIP1. Data are from two biological replicates. TSS, transcription start site; TES, transcription end site. "-2 kb" and "2 kb" represent the 2-kb regions upstream of TSS and downstream of TES, respectively. (**B**) Scatter plot showing the correlation of HDA19-Flag ChIP-seq signals in *hda19* and *hda19hdip1/2/3* mutant backgrounds at the target genes shared by HDA19 and HDIP1. Data are based on two biological replicates. The Pearson correlation coefficient (R) and the associated significance (P values) are shown. P values were determined by two-sided Pearson correlation test. (**C**) The enrichment of HDA19-Flag at the *PYL8*, *SnRK2.3* and *SARD1* loci determined by ChIP-qPCR in *hda19* and *hda19hdip1/2/3* mutant backgrounds. The chromatin from MCF7 cells (human) with Flag-ESR1 was added to the target chromatin and used as a spike-in control. *GREB1* represents the positive locus of the spike-in genome. Bar are means of three independent biological replicates ± SD. (**D**) The expression levels of *HDIP1-Flag* in the wild-type and *hda19* mutant backgrounds as determined by quantitative RT-PCR. Data are from two biological replicates. (**E**) Determination of the expression of the *HDIP1-Flag* in the wild-type and *hda19* mutant backgrounds by western blot analysis. Ponceau S-stained ribosome proteins are shown as a loading control. Source data are available online for this figure.

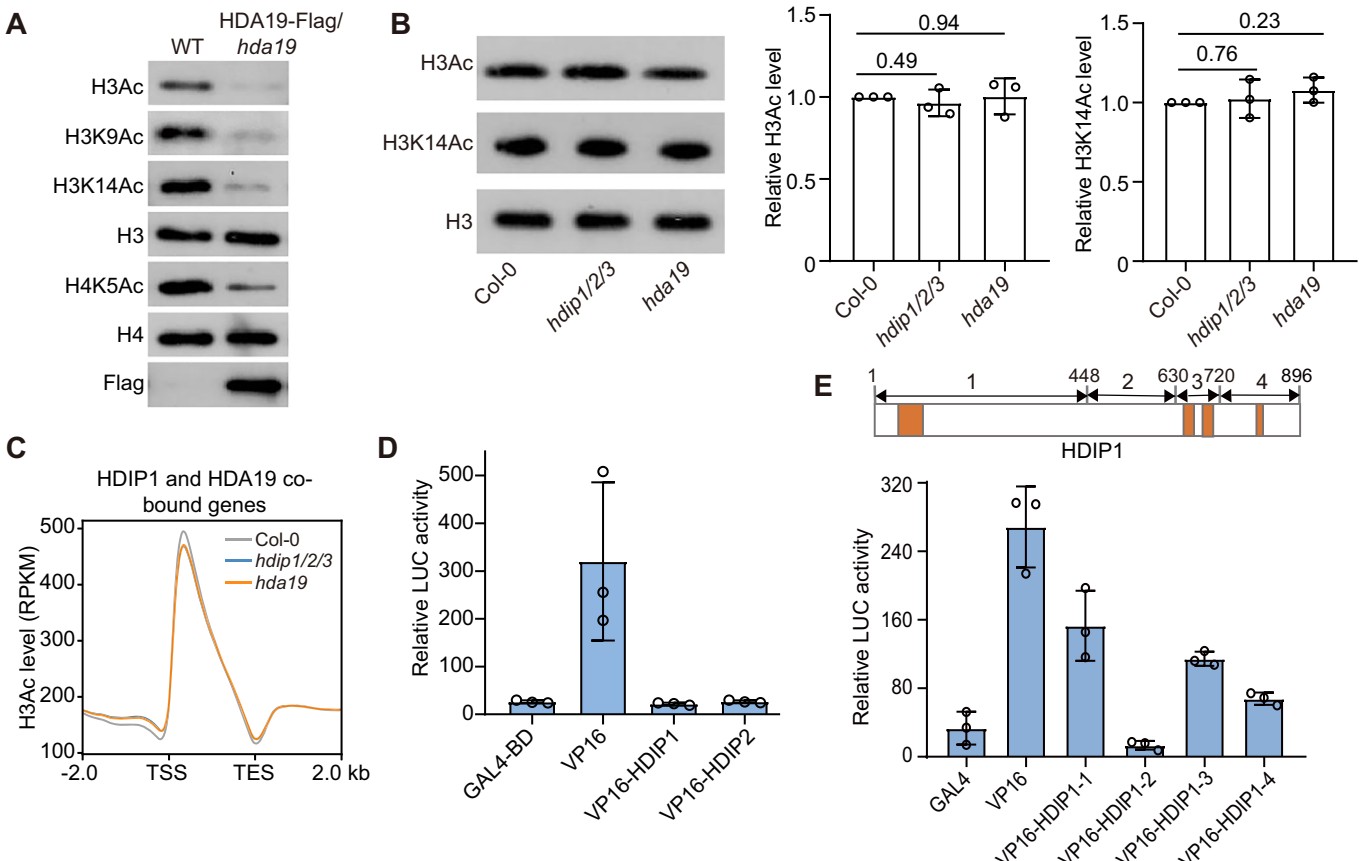

**Figure EV4. The roles of HDA19 and HDIP1/2/3 in histone deacetylation and transcriptional repression.**

(A) Determination of the histone deacetylase activity of the HDA19-containing complex. The HDA19-containing complex was purified from *HDA19-Flag* transgenic plants, and incubated with free histone substrates from calf thymus for the histone deacetylation assay. Immunoblots signals were detected by anti-H3Ac, anti-H3K9Ac, anti-H3K14Ac, and H4K5Ac antibodies. (B) Detection of the effect of *hda19* and *hdip1/2/3* mutants on H3 acetylation by western blot analysis. Quantifications of the H3Ac and H3K14Ac are shown in bar graphs. *P* values were determined by two-tailed Student's *t* test. (C) Metaplots showing the average distribution of H3Ac at HDIP1 and HDA19 shared target genes in Col-0, *hdip1/2/3*, and *hda19* mutants. (D) The transcriptional repression capacity of HDIP1 and HDIP2 as determined by the LUC reporter assay. Values are means ± SD of three biological replicates. (E) Determination of transcriptional repression ability of truncated HDIP1 by the LUC reporter assay. Diagrams represent truncated versions of the HDIP1 protein (upper). Truncated HDIP1-1 (1–448 aa), HDIP1-2 (449–630 aa), HDIP1-3 (631–720 aa), and HDIP1-4 (721–896 aa) were fused with the VP16 activation domain driven by the *CaMV 35S* promoter. Values are means ± SD of three independent biological replicates. Source data are available online for this figure.

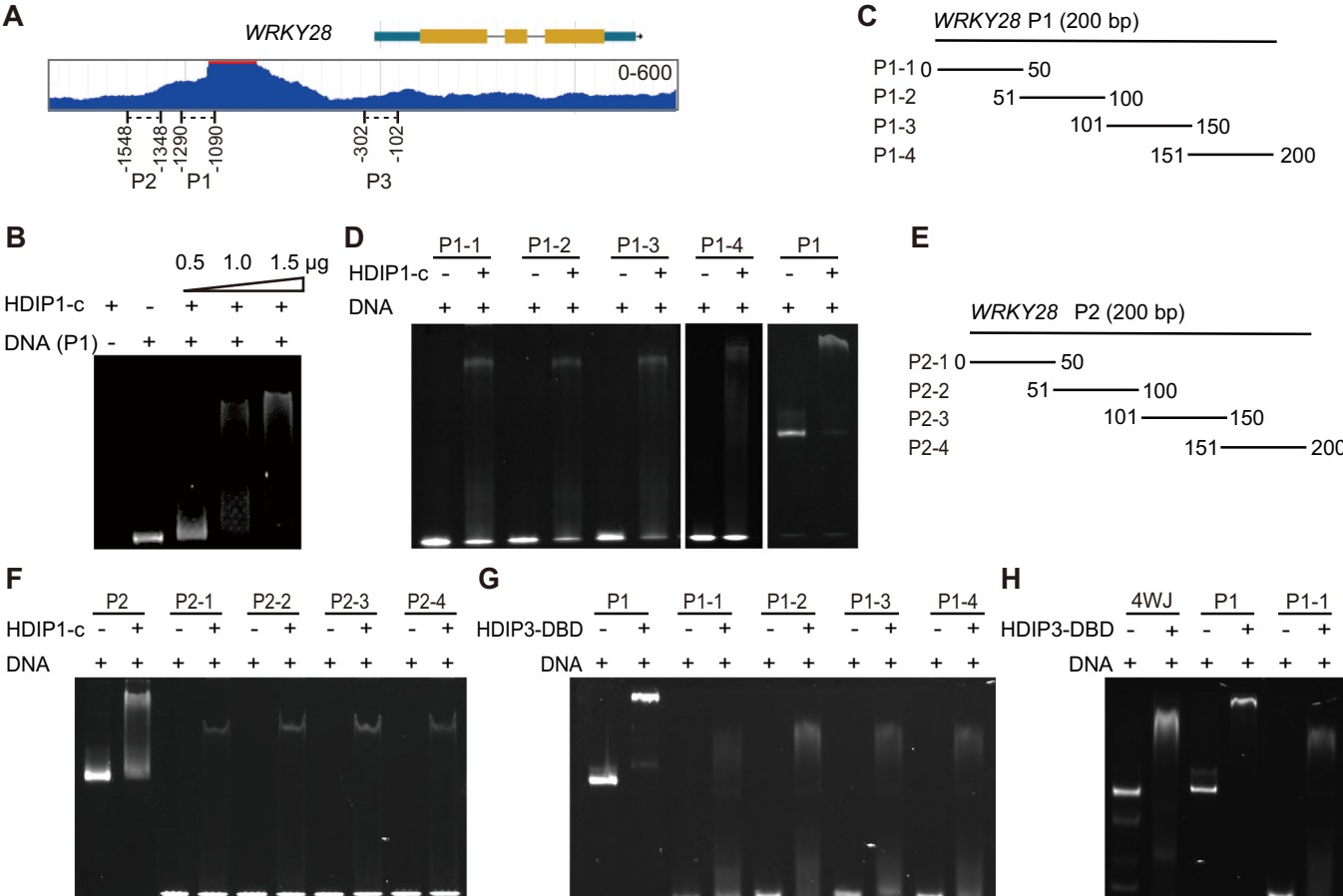

**Figure EV5.** Determination of the binding ability of HDIP1 and HDIP3 with various DNA probes by EMSA.

(A) The HDIP1 ChIP-seq signal across the *WRKY28* gene and the locations of the 200-bp P1, P2, and P3 DNA probes. (B) The binding of HDIP1 to the 200-bp double-stranded probe P1 as determined by EMSA. Increasing amounts of the HDIP1-c protein were used in the binding reaction mixture. (C) Schematic representations of the complete 200-bp DNA probe P1 and its truncated 50-bp versions: P1-1, P1-2, P1-3, and P1-4. (D) The binding of HDIP1-c to the 200-bp DNA probe P1 and its truncated 50-bp derivatives as determined by EMSA. (E) Schematic representations of the 200-bp DNA probe P2 and its truncated 50-bp versions: P2-1, P2-2, P2-3, and P2-4. (F) The binding of HDIP1-c to the 200-bp DNA probe P2 and its truncated 50-bp derivatives as determined by EMSA. (G) The binding of HDIP3-DBD (806–930 aa) to the 200-bp DNA probe P1 and its truncated 50-bp derivatives as determined by EMSA. (H) Determination of the binding ability of HDIP3-DBD with 4WJ DNA by EMSA. The 50-bp P1-1 probe and the 200-bp P1 probe were used as controls. Source data are available online for this figure.

