## [Peer Review File · The EMBO Journal]

Essential angiosperm-specific subunits of HDA19 histone deacetylase complexes in Arabidopsis

Na Liu, Jia-Xin Li, Dan-Yang Yuan, Yin-Na Su, Pei Zhang, Qi Wang, Xiao-Min Su, Lin Li, Haitao Li, She Chen, and Xin-Jian He

Corresponding author(s): Xin-Jian He (hexinjian@nibs.ac.cn)

Review Timeline:

Submission Date:	2nd Sep 24
Editorial Decision:	7th Oct 24
Revision Received:	21st Jan 25
Editorial Decision:	5th Feb 25
Revision Received:	14th Feb 25
Accepted:	31st Mar 25

Editor: William Teale

Transaction Report:

Dear Dr. He,

Thank you again for the submission of your manuscript entitled "Essential angiosperm-specific subunits of HDA19 histone deacetylase complexes in Arabidopsis" and for your patience during the review process. We have now received the reports from the referees, which I copy below.

As you can see from their comments, while all referees appreciated the relevance and quality of the work you present, Referee #2 and Referee #3 request a finer resolution be provided, especially around the molecular mechanism of HDIP-mediated repression tested in Figure 5. This (as well as other clarifications) will require your attention before your manuscript can be published in The EMBO Journal.

Based on the overall interest expressed in the reports, however, I would like to invite you to address the comments of all referees in a revised version of the manuscript. I should add that it is The EMBO Journal policy to allow only a single major round of revision and that it is therefore important to resolve the main concerns at this stage. I believe the concerns of the referees are reasonable and addressable, but please contact me if you have any questions, need further input on the referee comments or if you anticipate any problems in addressing any of their points. I am available via Zoom to discuss the reports with you at any time; let me know if you would like to go through them together. Please, follow the instructions below when preparing your manuscript for resubmission.

I would also like to point out that as a matter of policy, competing manuscripts published during this period will not be taken into consideration in our assessment of the novelty presented by your study ("scooping" protection). We have extended this 'scooping protection policy' beyond the usual 3 month revision timeline to cover the period required for a full revision to address the essential experimental issues. Please contact me if you see a paper with related content published elsewhere to discuss the appropriate course of action.

Again, please contact me at any time during revision if you need any help or have further questions.

Thank you very much again for the opportunity to consider your work for publication. I look forward to your revision.

Best regards,

William

William Teale, Ph.D.
Editor
The EMBO Journal

When submitting your revised manuscript, please carefully review the instructions below and include the following items:

- 1) a .docx formatted version of the manuscript text (including legends for main figures, EV figures and tables). Please make sure that the changes are highlighted to be clearly visible.
- 2) individual production quality figure files as .eps, .tif, .jpg (one file per figure).
- 3) a .docx formatted letter INCLUDING the reviewers' reports and your detailed point-by-point response to their comments. As part of the EMBO Press transparent editorial process, the point-by-point response is part of the Review Process File (RPF), which will be published alongside your paper.
- 4) a complete author checklist, which you can download from our author guidelines ([https://wol-prod-cdn.literatumonline.com/pb-assets/embo-site/Author Checklist%20-%20EMBO%20J-1561436015657.xlsx](https://wol-prod-cdn.literatumonline.com/pb-assets/embo-site/Author%20Checklist%20-%20EMBO%20J-1561436015657.xlsx)). Please insert information in the checklist that is also reflected in the manuscript. The completed author checklist will also be part of the RPF.
- 5) Please note that all corresponding authors are required to supply an ORCID ID for their name upon submission of a revised manuscript.
- 6) We require a 'Data Availability' section after the Materials and Methods. Before submitting your revision, primary datasets

produced in this study need to be deposited in an appropriate public database, and the accession numbers and database listed under 'Data Availability'. Please remember to provide a reviewer password if the datasets are not yet public (see <https://www.embopress.org/page/journal/14602075/authorguide#datadeposition>). If no data deposition in external databases is needed for this paper, please then state in this section: This study includes no data deposited in external repositories. Note that the Data Availability Section is restricted to new primary data that are part of this study.

Note - All links should resolve to a page where the data can be accessed.

8) For data quantification: please specify the name of the statistical test used to generate error bars and P values, the number (n) of independent experiments (specify technical or biological replicates) underlying each data point and the test used to calculate p-values in each figure legend. The figure legends should contain a basic description of n, P and the test applied. Graphs must include a description of the bars and the error bars (s.d., s.e.m.).

9) We would also encourage you to include the source data for figure panels that show essential data. Numerical data can be provided as individual .xls or .csv files (including a tab describing the data). For 'blots' or microscopy, uncropped images should be submitted (using a zip archive or a single pdf per main figure if multiple images need to be supplied for one panel). Additional information on source data and instruction on how to label the files are available at .

10) We replaced Supplementary Information with Expanded View (EV) Figures and Tables that are collapsible/expandable online (see examples in <https://www.embopress.org/doi/10.15252/embj.201695874>). A maximum of 5 EV Figures can be typeset. EV Figures should be cited as 'Figure EV1, Figure EV2" etc. in the text and their respective legends should be included in the main text after the legends of regular figures.

12) Our journal encourages inclusion of *data citations in the reference list* to directly cite datasets that were re-used and obtained from public databases. Data citations in the article text are distinct from normal bibliographical citations and should directly link to the database records from which the data can be accessed. In the main text, data citations are formatted as follows: "Data ref: Smith et al, 2001" or "Data ref: NCBI Sequence Read Archive PRJNA342805, 2017". In the Reference list, data citations must be labeled with "[DATASET]". A data reference must provide the database name, accession number/identifiers and a resolvable link to the landing page from which the data can be accessed at the end of the reference. Further instructions are available at .

13) In order to increase the reproducibility and reach of your work, The EMBO Journal includes a table of reagents that were used in the study. Please provide this along with your revisions.

Further instructions for preparing your revised manuscript:

We realize that it is difficult to revise to a specific deadline. In the interest of protecting the conceptual advance provided by the work, we recommend a revision within 3 months (5th Jan 2025). Please discuss the revision progress ahead of this time with the editor if you require more time to complete the revisions. Use the link below to submit your revision:

Referee #1:

By using affinity purification combined with mass spectrometry (AP-MS), three angiosperm-specific HDA19-interacting homologous proteins, HDIP1/2/3 were identified. HDIP1/2/3 interact with HDA19, SNLs, HDC1, and MS11, but not with HDA6, thus specifically contributing to the assembly of HDA19-containing histone deacetylase complexes in Arabidopsis. The loss-of-function mutants of HDIP1/2/3 and HDA19 exhibit highly similar defects in both plant development and ABA sensitivity. Additionally, HDIP1/2/3 and HDA19 co-localize on chromatin and co-regulate transcriptional repression. This study uncovered previously unknown histone deacetylase complex subunits and highlighted an angiosperm-specific feature of the SIN3-type histone deacetylase complex, which is crucial for plant development and stress responses.

Major points:

1. Given the fact that HDIP1, HDIP2, and HDIP3 (HDIP1/2/3) are new identified components of the HDAC complexes, the authors may want to investigate their expression patterns and subcellular localization. In addition, whether HDIP1/2/3 mutations affect global acetylation levels of Arabidopsis also need to be analyzed using western blot.
2. It is important to investigate the genetic interaction between HDIP1/2/3 and HDA19. The authors also need to analyze the mutant phenotype defect in both HDIP1/2/3 and HDA19 such as the *hdip1/2/3 hda19* mutant in Figure 1 and Figure 7.
3. Line 235 - "we also conducted ChIP-seq for HDA19-Flag expressed in the *hda19* and *hda19 hdip1/2/3* mutant backgrounds, and found the enrichment level of HDA19 was not affected by the *hdip1/2/3* mutation (Appendix Figure S8A, S8B), indicating that HDIP1/2/3 are dispensable for the association of HDA19 with chromatin".

Is the enrichment level of HDIP1 affected by the *hda19* mutation? Addressing this questions could provide additional information about the functional interaction between HDIP1/2/3 and HDA19.

4. Line 295 "While the effector construct expressing the transcription factor VP16 activated the LUC activity significantly, the

fusion of HDIP1 or HDIP2 to VP16 substantially inhibited the activity (Figure 5I), indicating the role of HDIP1/2 in transcriptional repression".

The fusion of HDIP1 or HDIP2 to VP16 may affect the confirmation and/or folding of VP16. To prove HDIP1/HDIP2 alone can act as a transcription repressor, the authors may want to co-transform the fusion of HDIP1/HDIP2 to GAL4-BD with a strong promoter with GAL4-UAS

5. Line 350 "To investigate whether the DNA-binding domain (DBD) of HDIP1 functions in Arabidopsis plants, we generated a construct expressing DBD-deleted HDIP1 (HDIP1-DBD) and transformed it into the *hdip1/2/3* mutant".

Is the DNA-binding domain (DBD) of HDIP1 also conserved in HDIP2/3? Can HDIP2/3 also bind to DNA. The conservation of DBD in HDIP2/3 and the binding ability of HDIP2/3 to DNA can further support the authors' claim that DNA-binding ability of HDIPs is important.

Minor points:

1. Line 213 - "Gene Ontology (GO) analysis revealed that the upregulated DEGs in both *hdip1/2/3* and *hda19* mutants are related to biotic and abiotic stress responses, the abscisic acid (ABA) signaling pathway, and salicylic acid (SA)-mediated plant immunity (Figure 3D)".

Since *hdip1/2/3* and *hda19* mutant exhibited severe developmental defects, are those up-regulated or down-regulated DEGs in both *hdip1/2/3* and *hda19* mutants related to developmental regulation?

2. Line 250 - "we individually assessed the enrichment of HDIP1 and HDA19 at three groups of genes, including HDIP1 and HDA19 shared target genes (group A), the overlap between group A genes and co-upregulated DEGs (defined as group B)".

The group B genes might be the direct target genes co-regulated by HDIP1 and HDA19. I suggest that the authors can list the group genes in a table and do Gene Ontology (GO) analysis of these genes.

3. Line 313 - "By using four truncated versions of the HDIP1 protein (HDIP1-a, HDIP1-b, HDIP1-c and HDIP1-d) (Figure 6A), our EMSA results showed that HDIP1-c, but not the other truncated forms of HDIP1, is capable of binding to all the three approximately 200-bp double-stranded DNA probes (P1, P2, and P3)",

Why the full length HDIP1 was not used in this experiment?

4. Figure 7G. H3Ac ChIP-seq levels of the ABA signaling pathway genes in the wild type, *hda19*, and *hdip1/2/3* mutants. Data are based on two independent biological replicates.

Please also show the genome browser view of H3Ac ChIP-seq signals.

Referee #2:

The manuscript by Liu et al. identifies HDA19 complex-specific HDIP proteins and investigates their function through various genomic and biochemical approaches. The authors initially identified these proteins using proteomics and then performed a comprehensive analysis to determine the relevant regions that interact with other HDA19 complex subunits. They also generated HDIP mutant combinations, demonstrating phenotypic similarities with HDA19 mutants under normal conditions and in response to stress/ABA. This was further supported by similar expression changes revealed through RNA-seq, overlapping genomic targets identified by ChIP-seq, and changes in histone acetylation, also assessed by ChIP-seq. Additionally, the authors demonstrated the DNA-binding activity of HDIP.

This is the first manuscript to report on the HDIP phenotype, its genomic localization, and its connection to the HDA19 complex and histone acetylation. Moreover, the identification of DNA binding to four-way junction DNA structures and nucleosomes provides valuable insights into the functioning of histone deacetylation complexes in plants.

The experiments presented are of high quality, as are the text and figures. However, my main criticism pertains to the claim that HDIP and HDA19 can regulate gene expression in a histone acetylation-independent manner. See comment below regarding Figures 5H-J and Figure 5/7.

Here are some comments and suggestions for the authors to consider to improve the manuscript:

- Figure 1: Distinct sets of SNLs are detected in different HDIP IPs. For example, HDIP1 IPs show SNL3, 4, 5, 6, while HDIP2

detects all of them. Is this due to technical issues? How many replicates were done for these experiments? This variability makes it difficult to conclude whether the deletion of alpha2/3 in HDIP1 (Figure 1F) affects its ability to interact with specific SNLs.

- Figure 1F: The figure legend mentions that the delta C deletion corresponds to amino acids 631-720. Is this a typo? If not, removing this C-terminal region, which contains an NLS, might prevent nuclear access and interaction with other subunits. This should be tested via microscopy or western blotting after cell fractionation, separating the nucleus and cytoplasm.

- Figure 4B: Are the HDA19-unique targets not bound by HDIP1 at all? A heatmap and/or metaplot showing HDIP1 and HDA19 binding over these unique HDA19 targets would be very informative.

- Appendix Figure S6: It is shown that the hdip1/2/3 triple mutant is stronger than the hdip1/3 double mutant. Why didn't the authors test the hdip1/2 and hdip2/3 combinations? These combinations may be sufficient to recapitulate the triple mutant phenotype.

- Appendix Figure S8: This figure shows no effect of HDIP on HDA19 chromatin binding. If this effect occurs equally across all HDA19 binding sites, the ChIP-seq analysis may mask differences due to normalization issues. These experiments would strongly benefit from spike-in controls using chromatin from a different organism expressing FLAG-tagged proteins to normalize the results. I suggest the authors repeat the ChIP-seq experiment with a spike-in control, or at least perform ChIP-qPCR on selected loci.

- Figure 5: Could the authors present metaplots of histone acetylation ChIP-seq data from WT, hdip, and hda19 mutants over distinct clusters (e.g., HDIP-HDA19 common targets)? This would help identify genic regions where acetylation increases (or decreases). Additionally, representing this data alongside HDIP and HDA19 ChIP-seq signals could be informative. Based on the HDIP and HDA19 ChIP-seq signals, one might expect stronger effects over the 5' regions.

- Figure 5H-J: The authors conclude that HDIP has an acetylation-independent effect on transcription. I disagree with this interpretation because multiple factors may be at play. For instance, HDA19 alone may not be sufficient to recruit the rest of the complex and achieve full deacetylation activity, while adding HDIP might allow for full complex reconstitution. A straightforward experiment to support the authors' conclusion would be to show that VP16-HDIP1 does not trigger deacetylation, using ChIP-qPCR as validation.

- Figure 5/7: The data does not convincingly support the conclusion that HDIP exerts an acetylation-independent effect on gene expression. The authors only tested a pan-H3 acetyl antibody, but does this antibody detect all possible H3 acetylation sites? Furthermore, can the authors rule out that the HDA19 complex acts only on H3? It is possible that HDA19 also affects other histones, such as H4. For example, previous reports have shown that SIN3 and RPD3 deacetylases can target H4. If so, hda19 and hdip mutants might show acetylation changes (either on H4 or alternative H3 sites) linked to the upregulation of ABA-related genes. To support the authors' conclusions, ChIP-seq with additional histone acetylation antibodies should be performed in hda19 and hdip mutants. Ideally, in vitro experiments should identify the exact modified histone residues.

- Figure 6: The authors claim that HDIP prefers longer DNA fragments over shorter ones but only test this with one probe. To make this conclusion more robust, it would be helpful to show similar results using probe 2 and/or 3.

- Figure 6H: The authors demonstrate that HDIP1 interacts with nucleosomes. How do they interpret the observation that HDIP1 and HDA19 ChIP-seq profiles resemble binding to open chromatin (similar to ATAC-seq) rather than genic nucleosomes?

- Figure 6I: This panel shows the in vivo relevance of the DNA-binding domain (DBD) in HDIP function. Do the authors believe this is due to inefficient DNA binding by the complex? A ChIP experiment using the deleted HDIP version could provide insights into this question.

Minor changes:

- Line 165: Correct "Appendix figure S4A-S4D."

- Line 229: HDA19 targets total should read 13,366.

- Line 259: "Associated" should be corrected.

- Line 416: "MIS1" requires correction.

- A brief description of the nucleosome assembly protocol and the criteria for selecting the DNA sequence should be added.

- The reference to Panday & Grove, 2016 does not seem appropriate for discussing four-way junction DNA and nucleosomes.

- Lines 381-396 appear to belong in the Discussion section rather than Results.

Referee #3:

The authors have conducted extensive work and identified HDA19 interactors-HDIP1, HDIP2, and HDIP3-through IP-mass spectrometry. Genetic studies showed that mutations in HDIP1, HDIP2, and HDIP3 phenocopy mutations in HDA19. Additionally, they demonstrated that HDA19, along with HDIP1, HDIP2, and HDIP3, co-occupies chromatin and functions together to repress gene transcription. While this is an interesting discovery of new components, the excitement is somewhat tempered by the lack of novel mechanisms involved in this regulatory process.

Here I list some main concerns:

1. IP-mass spectrometry results showed that HDA19 interacts with HDIP1, HDIP2, and HDIP3. However, the in vitro pull-down assay was only performed with HDIP1. Notably, the sequence similarity among the three HDIPs is not particularly high, especially for HDIP3 (Fig S1A). It is important to determine whether all three HDIPs directly interact with HDA19 or if they form a protein complex in which only one of them mediates the interaction with HDA19.

Additionally, although HDA6 was not pulled down by HDIP1, HDIP2, or HDIP3 in their IP-mass assays, it was detected in the SNL5 IP-mass assay, and SNL5 interacts with HDIP1. This suggests the possibility that HDA6 might still interact with HDIP1 indirectly. A second thought of the conclusion that HDIP1 specifically interacts with HDA19 should be given.

2. Fig 5. The LUC reporter system is designed to detect transcriptional activation or repression, depending on whether effectors bind to the promoter (in this case, 5xUAS) to either activate or repress LUC gene expression. It is unclear how HDA19 functions as a repressor by itself in this system. Additionally, did the authors examine histone acetylation levels at the 5xUAS site? Does HDA19 repress LUC gene expression through histone deacetylation?

3. Fig 6. As in fig6I showed that mutation in HDIP1 cause variations in bolting time. Would mutations in HDIP1 affect its binding in vivo lead to the phenotype? or influence the interaction with its co factors?

Minor: Fig7. It is unclear why figure 7 is necessary for this manuscript.

Referee #1:

By using affinity purification combined with mass spectrometry (AP-MS), three angiosperm-specific HDA19-interacting homologous proteins, HDIP1/2/3 were identified. HDIP1/2/3 interact with HDA19, SNLs, HDC1, and MSI1, but not with HDA6, thus specifically contributing to the assembly of HDA19-containing histone deacetylase complexes in Arabidopsis. The loss-of-function mutants of HDIP1/2/3 and HDA19 exhibit highly similar defects in both plant development and ABA sensitivity. Additionally, HDIP1/2/3 and HDA19 co-localize on chromatin and co-regulate transcriptional repression. This study uncovered previously unknown histone deacetylase complex subunits and highlighted an angiosperm-specific feature of the SIN3-type histone deacetylase complex, which is crucial for plant development and stress responses.

Response: Thank you very much for your positive comments. All the comments and suggestions have been point-by-point addressed in the revised manuscript.

Major points:

1. Given the fact that HDIP1, HDIP2, and HDIP3 (HDIP1/2/3) are new identified components of the HDAC complexes, the authors may want to investigate their expression patterns and subcellular localization. In addition, whether HDIP1/2/3 mutations affect global acetylation levels of Arabidopsis also need to be analyzed using western blot.

Response: As suggested, we have analyzed the expression patterns of *HDIP1*, *HDIP2*, and *HDIP3* analysis based on ATHENA databases (http://athena.proteomics.wzw.tum.de:5002/master_arabidopsisshiny/) in Appendix Fig S1. Additionally, we performed nuclear-cytoplasmic fractionation using *HDIP1-Flag*, *HDIP3-Flag*, and *HDIP1-AC-Flag* transgenic plants, followed by western blot analysis. The results indicated that HDIP1 and HDIP3 were localized in the nucleus, and the deletion of the C-terminal domain did not affect the location of HDIP1 in the nucleus (Appendix Fig S6). To determine the effect of *hdip1/2/3* on the global histone acetylation level, we performed western blot analysis to examine the histone acetylation level in the *hdip1/2/3* mutant as well as in the *hda19* mutant and the wild type, indicating that the histone acetylation level was not affected in the *hdip1/2/3* or *hda19* mutants compared to in the wild type

(Appendix Fig S13B).

2. It is important to investigate the genetic interaction between HDIP1/2/3 and HDA19. The authors also need to analyze the mutant phenotype defect in both HDIP1/2/3 and HDA19 such as the *hdip1/2/3hda19* mutant in Figure 1 and Figure 7.

Response: Thank you for the valuable suggestion. We agree with you that analyzing the phenotype of the *hdip1/2/3hda19* quadruple mutant will be helpful for understanding the genetic relationship between *hdip1/2/3* and *hda19*. Due of time limitation of the revision, we cannot provide the phenotypic data for the *hdip1/2/3hda19* mutant in the current version. We feel that the results shown in the current version are sufficient for supporting the conclusion that HDIP1/2/3 are angiosperm-specific subunits of HDA19 histone deacetylase complexes in Arabidopsis. We will further investigate the role on HDIP1/2/3 in the following study.

3. Line 235 - "we also conducted ChIP-seq for HDA19-Flag expressed in the *hda19* and *hda19 hdip1/2/3* mutant backgrounds, and found the enrichment level of HDA19 was not affected by the *hdip1/2/3* mutation (Appendix Figure S8A, S8B), indicating that HDIP1/2/3 are dispensable for the association of HDA19 with chromatin".

Is the enrichment level of HDIP1 affected by the *hda19* mutation? Addressing this questions could provide additional information about the functional interaction between HDIP1/2/3 and HDA19.

Response: To detect the enrichment level of HDIP1 affected by *had19* mutant, we transformed the HDIP1-Flag transgene into *hda19* mutant as well as in the wild-type background. Interestingly, we found that while the transcript level of *HDIP1-Flag* transgene in the *hda19* mutant was similar with that in the wild-type background (Appendix Fig S11D), the protein levels of the transgene were detected in the wild-type background but not in the *hda19* mutant (Appendix Fig S11E). This result indicates that HDA19 is essential for maintaining the protein level of HDIP1, supporting the notion that HDIP1/2/3 are subunits of HDA19 histone deacetylase complexes in Arabidopsis plants.

4. Line 295 "While the effector construct expressing the transcription factor VP16 activated the LUC

activity significantly, the fusion of HDIP1 or HDIP2 to VP16 substantially inhibited the activity (Figure 5I), indicating the role of HDIP1/2 in transcriptional repression".

The fusion of HDIP1 or HDIP2 to VP16 may affect the confirmation and/or folding of VP16. To prove HDIP1/HDIP2 alone can act as a transcription repressor, the authors may want to co-transform the fusion of HDIP1/HDIP2 to GAL4-BD with a strong promoter with GAL4-UAS.

Response: Thanks for your suggestion. Considering that the fusion of HDIP1 to VP16 may affect the conformation or folding of VP16, we further generated a series of truncated versions of HDIP1 to VP16, and then determined their effects on the LUC activity. The result indicated that the truncated form HDIP1-2 exhibits a major repressive effect on transcription while the other truncated forms have minor effects (Appendix Fig S13E), suggesting that HDIP1 has a specific role in transcription repression.

5. Line 350 "To investigate whether the DNA-binding domain (DBD) of HDIP1 functions in Arabidopsis plants, we generated a construct expressing DBD-deleted HDIP1 (HDIP1-DBD) and transformed it into the *hdip1/2/3* mutant".

Is the DNA-binding domain (DBD) of HDIP1 also conserved in HDIP2/3? Can HDIP2/3 also bind to DNA. The conservation of DBD in HDIP2/3 and the binding ability of HDIP2/3 to DNA can further support the authors' claim that DNA-binding ability of HDIPs is important.

Response: In terms of sequence conservation, the DNA-binding domain (DBD) of HDIP1 is highly conserved in HDIP2 and HDIP3 (Appendix Fig S2A). To determine whether the corresponding domain in HDIP3 can also bind to DNA, we purified HDIP3-DBD (806-930 aa) protein and determined its DNA-binding ability. Our EMSA result indicated that HDIP3-DBD was also capable of binding to DNA. Similarly, the ability of HDIP3-DBD for long DNA (200 bp) was stronger than that for short DNA (50 bp) (Appendix Fig S14G). Additionally, like HDIP1, HDIP3 can also efficiently bind to the four-way junction (Appendix Fig S14H). Therefore, the binding preference for long DNA and four-way junction is similar between HDIP1 and HDIP3.

Minor points:

1. Line 213 - "Gene Ontology (GO) analysis revealed that the upregulated DEGs in both *hdip1/2/3* and *hda19* mutants are related to biotic and abiotic stress responses, the abscisic acid (ABA) signaling pathway, and salicylic acid (SA)-mediated plant immunity (Figure 3D)".

Since *hdip1/2/3* and *hda19* mutant exhibited severe developmental defects, are those up-regulated or down-regulated DEGs in both *hdip1/2/3* and *hda19* mutants related to developmental regulation?

Response: Our RNA-seq data have indicated that the upregulated DEGs in both *hdip1/2/3* and *hda19* mutants are related to biotic and abiotic stress responses (Fig 3E). Previous studies have suggested that increased expression of biotic and abiotic responsive genes can suppress the normal growth and development (van Verk et al., 2011; Wang et al., 2011). Therefore, we speculated that the developmental defects in the *hdip1/2/3* and *hda19* mutants are caused by those up-regulated stress-responsive genes.

2. Line 250 - "we individually assessed the enrichment of HDIP1 and HDA19 at three groups of genes, including HDIP1 and HDA19 shared target genes (group A), the overlap between group A genes and co-upregulated DEGs (defined as group B)".

The group B genes might be the direct target genes co-regulated by HDIP1 and HDA19. I suggest that the authors can list the group genes in a table and do Gene Ontology (GO) analysis of these genes.

Response: As suggested, we have listed the genes (group B) in Dataset EV4 and analyzed Gene Ontology (GO) enrichment of group B genes. The results indicated that these genes involved in abscisic acid (ABA) response, salicylic acid (SA) response and defense response to bacteria (Appendix Fig S12), confirming that HDIP1 and HDA19 form a histone deacetylase complex to suppress the expression of stress-responsive genes.

3. Line 313 - "By using four truncated versions of the HDIP1 protein (HDIP1-a, HDIP1-b, HDIP1-c and HDIP1-d) (Figure 6A), our EMSA results showed that HDIP1-c, but not the other truncated forms of HDIP1, is capable of binding to all the three approximately 200-bp double-stranded DNA probes (P1, P2, and P3)",

Why the full length HDIP1 was not used in this experiment?

Response: We initially tried to obtain the full length of HDIP1 for this experiment, but the HDIP1 protein was not successfully purified in *Escherichia coli* or yeast cells. Therefore, we used a series of truncated versions of HDIP1 in this experiment. Because these truncated versions of HDIP1 covered the full-length HDIP1, the experiment can provide more details for its binding ability.

4. Figure 7G. H3Ac ChIP-seq levels of the ABA signaling pathway genes in the wild type, *hda19*, and *hdip1/2/3* mutants. Data are based on two independent biological replicates.

Please also show the genome browser view of H3Ac ChIP-seq signals.

Response: As suggested, we have showed the genome browser view of H3Ac ChIP-seq signals in the Appendix Fig S16.

References

van Verk MC, Bol JF & Linthorst HJ (2011) WRKY Transcription Factors Involved in Activation of SA Biosynthesis Genes. *BMC Plant Biology* 11: 89

Wang L, Tsuda K, Truman W, Sato M, Nguyen LV, Katagiri F & Glazebrook J (2011) CBP60g and SARD1 play partially redundant critical roles in salicylic acid signaling. *The Plant Journal* 67: 1029–1041

Referee #2:

The manuscript by Liu et al. identifies HDA19 complex-specific HDIP proteins and investigates their function through various genomic and biochemical approaches. The authors initially identified these proteins using proteomics and then performed a comprehensive analysis to determine the relevant regions that interact with other HDA19 complex subunits. They also generated HDIP mutant combinations, demonstrating phenotypic similarities with HDA19 mutants under normal conditions and in response to stress/ABA. This was further supported by similar expression changes revealed through RNA-seq, overlapping genomic targets identified by ChIP-seq, and changes in histone acetylation, also assessed by ChIP-seq. Additionally, the authors demonstrated the DNA-binding

activity of HDIP.

This is the first manuscript to report on the HDIP phenotype, its genomic localization, and its connection to the HDA19 complex and histone acetylation. Moreover, the identification of DNA binding to four-way junction DNA structures and nucleosomes provides valuable insights into the functioning of histone deacetylation complexes in plants.

The experiments presented are of high quality, as are the text and figures. However, my main criticism pertains to the claim that HDIP and HDA19 can regulate gene expression in a histone acetylation-independent manner. See comment below regarding Figures 5H-J and Figure 5/7. Here are some comments and suggestions for the authors to consider to improve the manuscript:

Response: We appreciate the reviewer's insightful and helpful suggestion. We have addressed the comments point-by-point below.

- Figure 1: Distinct sets of SNLs are detected in different HDIP IPs. For example, HDIP1 IPs show SNL3, 4, 5, 6, while HDIP2 detects all of them. Is this due to technical issues? How many replicates were done for these experiments? This variability makes it difficult to conclude whether the deletion of alpha2/3 in HDIP1 (Figure 1F) affects its ability to interact with specific SNLs.

Response: Thanks for your suggestion. Distinct sets of SNLs is due to technical issues among different AP-MS experiments. SNL1, SNL2, and SNL4 were detected in the HDIP2 AP-MS result but not in the HDIP3 AP-MS result (Fig 1A). The protein levels of SNL1, SNL2, and SNL4 detected in the HDIP2 AP-MS result were lower than the protein levels of other SNL proteins (Fig 1A), supporting the notion that the failure in the detection of SNL1, SNL2, and SNL4 in the HDIP3 AP-MS result is caused by the technical limitation. At least two replicates were performed for these MS experiments, confirming the results shown in the manuscript. Our AP-MS data indicated that the deletion of alpha2/3 in HDIP1 can only weakly affected the interaction with SNLs, while the deletion of the C-terminal region can completely disrupt the interaction of HDIP1 with the HDA19 histone deacetylase complex (Fig 1F).

- Figure 1F: The figure legend mentions that the delta C deletion corresponds to amino acids 631-720. Is this a typo? If not, removing this C-terminal region, which contains an NLS, might prevent nuclear

access and interaction with other subunits. This should be tested via microscopy or western blotting after cell fractionation, separating the nucleus and cytoplasm.

Response: As suggested, we explored whether the deletion of the C-terminal region affects the subcellular localization of HDIP1 via nuclear-cytoplasmic fractionation followed by western blotting. The results indicated that HDIP1- Δ C was located in the nucleus as well as the full-length HDIP1 (Appendix Fig S6). We speculated that HDIP1 might contain an NLS, which is not predicted by the NLStradamus tool, or it was brought into the nucleus through interaction with other proteins.

- Figure 4B: Are the HDA19-unique targets not bound by HDIP1 at all? A heatmap and/or metaplot showing HDIP1 and HDA19 binding over these unique HDA19 targets would be very informative.

Response: Thanks for your suggestion. To determine whether the HDA19-unique targets are also bound by HDIP1, we analyzed the correlation of peaks between HDIP1 and HDA19 and found that they showed a high positive correlation (Appendix Fig S10A). Additionally, we drew a heatmap for HDIP1 and HDA19 over the HDA19-specific targets, indicating that HDIP1 was also present at the HDA19-specific target genes (Appendix Fig S10B). We found that the overall enrichment level of HDA19 was higher at HDA19 and HDIP1 co-target genes than at HDA19-specific target genes (Appendix Fig S10C). Due to the overall HDIP1 enrichment level was lower than the HDA19 enrichment level (Appendix Fig S10C), the failure in identifying the HDA19-specific genes as HDIP1 targets was caused by the technical limitation of the HDIP1 ChIP-seq experiment.

- Appendix Figure S6: It is shown that the *hdip1/2/3* triple mutant is stronger than the *hdip1/3* double mutant. Why didn't the authors test the *hdip1/2* and *hdip2/3* combinations? These combinations may be sufficient to recapitulate the triple mutant phenotype.

Response: As suggested, we generated all the *hdip* single mutants and mutant combinations, including *hdip1*, *hdip2*, *hdip3*, *hdip1/2*, *hdip1/3*, *hdip2/3*, and *hdip1/2/3*. As shown in Appendix Fig S8, *hdip1*, *hdip2* and *hdip3* single mutant have no obvious developmental phenotype; the *hdip2/3* and *hdip1/2* mutants showed similar morphological phenotype with the *hdip1/3* double mutant; the *hdip1/2/3* triple

mutant exhibited the strongest phenotype (Appendix Fig S8A-C). This analysis strongly suggests that HDIP1/2/3 function redundantly.

- Appendix Figure S8: This figure shows no effect of HDIP on HDA19 chromatin binding. If this effect occurs equally across all HDA19 binding sites, the ChIP-seq analysis may mask differences due to normalization issues. These experiments would strongly benefit from spike-in controls using chromatin from a different organism expressing FLAG-tagged proteins to normalize the results. I suggest the authors repeat the ChIP-seq experiment with a spike-in control, or at least perform ChIP-qPCR on selected loci.

Response: As suggested, we performed ChIP-qPCR of HDA19-Flag in *hda19* and *hda19hdip1/2/3* mutant backgrounds with the MCF7 cell (human) with Flag-ESR1 as a spike-in control. In this experiment, we added 25% heterologous spike-in chromatin from human to the Arabidopsis chromatin before immunoprecipitation to normalize the ChIP-qPCR results. As shown in Appendix Fig S11C, the binding of HDA19 to its target genes was not affected in *hda19hdip1/2/3* mutant background relative to the *hda19* mutant background. These results strongly suggest that the HDIP1/2/3 are not required for the binding of HDA19 to chromatin.

- Figure 5: Could the authors present metaplots of histone acetylation ChIP-seq data from WT, *hdip*, and *hda19* mutants over distinct clusters (e.g., HDIP-HDA19 common targets)? This would help identify genic regions where acetylation increases (or decreases). Additionally, representing this data alongside HDIP and HDA19 ChIP-seq signals could be informative. Based on the HDIP and HDA19 ChIP-seq signals, one might expect stronger effects over the 5' regions.

Response: As suggested, we used metaplots to show the histone acetylation signal of the HDIP1 and HDA19 co-targeted genes in the WT, *hdip1/2/3*, and *hda19* mutants. The results showed that the overall histone acetylation levels were not significantly affected in the *hdip1/2/3* or *hda19* mutants compared with the wild type (Appendix Fig S13C), which is consistent with the finding that only a small number of HDIP1 and HDA19 co-target genes showed an increased H3Ac level in *hdip1/2/3* (599/7871) and *hda19* (767/7871) (Fig 5C).

- Figure 5H-J: The authors conclude that HDIP has an acetylation-independent effect on transcription. I disagree with this interpretation because multiple factors may be at play. For instance, HDA19 alone may not be sufficient to recruit the rest of the complex and achieve full deacetylation activity, while adding HDIP might allow for full complex reconstitution. A straightforward experiment to support the authors' conclusion would be to show that VP16-HDIP1 does not trigger deacetylation, using ChIP-qPCR as validation.

Response: Thanks for the suggestion. As suggested, we performed ChIP-qPCR for VP16-HDIP1 to determine whether histone deacetylation is required for VP16-HDIP1-mediated transcriptional repression. As shown in Fig 5J, the H3Ac level of the LUC reporter gene was not affected by different effectors, even when the expression level of the reporter gene was significantly affected (Fig 5I and Appendix Fig S13D). This result supports the notion that HDIP1 can mediate transcriptional repression in a histone deacetylation-independent manner.

- Figure 5/7: The data does not convincingly support the conclusion that HDIP exerts an acetylation-independent effect on gene expression. The authors only tested a pan-H3 acetyl antibody, but does this antibody detect all possible H3 acetylation sites? Furthermore, can the authors rule out that the HDA19 complex acts only on H3? It is possible that HDA19 also affects other histones, such as H4. For example, previous reports have shown that SIN3 and RPD3 deacetylases can target H4. If so, *hda19* and *hdip* mutants might show acetylation changes (either on H4 or alternative H3 sites) linked to the upregulation of ABA-related genes. To support the authors' conclusions, ChIP-seq with additional histone acetylation antibodies should be performed in *hda19* and *hdip* mutants. Ideally, *in vitro* experiments should identify the exact modified histone residues.

Response: Thanks for your suggestion. We agree with the reviewer that HDA19 might remove acetylation from both histone H3 and H4. To test the idea, we utilized three additional acetyl-histone antibodies for detecting the histone deacetylation activity of HDA19, including H3K9Ac antibody, H3K14Ac antibody, and H4K5Ac antibody. By using purified HDA19 from *HDA19-Flag* transgenic plants, we performed *in vitro* histone deacetylation assays and found that HDA19 can mediate histone

deacetylation on free histone substrates at all tested acetylated lysine sites (Appendix Fig S13A). Therefore, it is reasonable to select a pan H3Ac antibody as a representative in the ChIP-seq experiment.

- Figure 6: The authors claim that HDIP prefers longer DNA fragments over shorter ones but only test this with one probe. To make this conclusion more robust, it would be helpful to show similar results using probe 2 and/or 3.

Response: As suggested, we tested the DNA-binding preference of HDIP1 by EMSA using both the full-length and truncated versions of probe 2 (Appendix Fig S14E). The results showed that the binding affinity of HDIP1 for the full-length probe 2 (200 bp) was markedly stronger than that for each truncated version of probe 2 (50 bp) (Appendix Fig S14F), confirming the notion that HDIP1 prefers longer DNA over shorter ones.

- Figure 6H: The authors demonstrate that HDIP1 interacts with nucleosomes. How do they interpret the observation that HDIP1 and HDA19 ChIP-seq profiles resemble binding to open chromatin (similar to ATAC-seq) rather than genic nucleosomes?

Response: Thanks for pointing out this. Our ChIP-seq results indicated that both the HDIP1 and HDA19 signals can form a sharp peak around the transcription start site, which is shortly followed by the +1 nucleosome in the gene body. Considering that HDIP1 has a strong affinity for four-way junction DNA, which mimics the DNA entry/exit sites of the nucleosome (Wang et al, 1998; Panday & Grove, 2016). Therefore, it is reasonable that HDIP1 and HDA19 are predominantly associated with the intersection between the promoter DNA and +1 nucleosome and form a peak around the transcription start site.

- Figure 6I: This panel shows the in vivo relevance of the DNA-binding domain (DBD) in HDIP function. Do the authors believe this is due to inefficient DNA binding by the complex? A ChIP experiment using the deleted HDIP version could provide insights into this question

Response: Thanks for your suggestion. Our ChIP-seq results have indicated that the *hdip1/2/3* mutation

did not affect the association of HDA19 with chromatin at the whole-genome level (Appendix Fig S11A-S11C), suggesting that the DNA-binding capacity of HDIP1 is not involved in the association of HDA19 histone deacetylase complex with chromatin. To validate the idea, we performed the ChIP-seq analysis using wild-type HDIP1-Flag and HDIP1- Δ DBD-Flag transgenic plants and found that the deletion of the DNA-binding domain (Δ DBD) did not show a major effect on the association of HDIP1 with chromatin (Appendix Fig S15D,S15E), confirming the notion that HDIP1 is not involved in the association of HDA19 with chromatin. We predict that the binding of HDIP1 to nucleosomal DNA functions at the downstream step of the HDA19 recruitment to chromatin, and likely facilitates histone deacetylation mediated by HDA19 on the nucleosome.

Minor changes:

- Line 165: Correct "Appendix figure S4A-S4D."

Response: We have corrected it as suggested.

- Line 229: HDA19 targets total should read 13,366.

Response: We have revised it as suggested.

- Line 259: "Associated" should be corrected.

Response: We have revised it as suggested.

- Line 416: "MIS1" requires correction.

Response: We have corrected it as suggested.

- A brief description of the nucleosome assembly protocol and the criteria for selecting the DNA sequence should be added.

Response: As suggested, we have supplemented the nucleosome assembly protocol in Materials and Methods.

- The reference to Panday & Grove, 2016 does not seem appropriate for discussing four-way junction DNA and nucleosomes.

Response: As suggested, we have deleted the reference the revised manuscript.

- Lines 381-396 appear to belong in the Discussion section rather than Results.

Response: As suggested, we have moved the text to the Discussion section in the revised manuscript.

Referee #3:

The authors have conducted extensive work and identified HDA19 interactors-HDIP1, HDIP2, and HDIP3-through IP-mass spectrometry. Genetic studies showed that mutations in HDIP1, HDIP2, and HDIP3 phenocopy mutations in HDA19. Additionally, they demonstrated that HDA19, along with HDIP1, HDIP2, and HDIP3, co-occupies chromatin and functions together to repress gene transcription. While this is an interesting discovery of new components, the excitement is somewhat tempered by the lack of novel mechanisms involved in this regulatory process.

Response: Thank you very much for your positive and constructive comments. We have point-by-point addressed these comments.

Here I list some main concerns:

1. IP-mass spectrometry results showed that HDA19 interacts with HDIP1, HDIP2, and HDIP3. However, the in vitro pull-down assay was only performed with HDIP1. Notably, the sequence similarity among the three HDIPs is not particularly high, especially for HDIP3 (Fig S1A). It is important to determine whether all three HDIPs directly interact with HDA19 or if they form a protein complex in which only one of them mediates the interaction with HDA19.

Additionally, although HDA6 was not pulled down by HDIP1, HDIP2, or HDIP3 in their IP-mass

assays, it was detected in the SNL5 IP-mass assay, and SNL5 interacts with HDIP1. This suggests the possibility that HDA6 might still interact with HDIP1 indirectly. A second thought of the conclusion that HDIP1 specifically interacts with HDA19 should be given.

Response: As suggested, we performed a pull-down assay to determine whether HDIP3 directly interacts with HDA19. The pull-down assay indicated that like the C-terminal domain of HDIP1, the C-terminal domain of HDIP3 (906-1005 aa) is also responsible for interaction with HDA19 (Appendix Fig S5C), suggesting that the HDIPs have a conserved ability to interact with HDA19.

We agree with the reviewer that it is confusing that HDIPs interact with SNLs and HDA19 but not with HDA6. Our AP-MS data indicated that while SNLs were co-purified with both HDA19-Flag and HDA6-Flag, HDIPs were specifically co-purified with HDA19-Flag but not with HDA6-Flag (Fig 1A). Vice versa, the AP-MS data also showed that HDA19 but not HDA6 was co-purified with HDIPs-Flag (Fig 1A). These results strongly suggest that HDIP1 specifically interact with HDA19 in Arabidopsis plants. Although the *in vitro* pull-down assay showed that SNL5 can directly interact with HDIP1, we predict that the interacting interface of SNL5 is possibly hidden in the HDA6 complex, resulting in the absence of HDIPs in the HDA6 complex. We have added the explanation to the revised manuscript.

2. Fig 5. The LUC reporter system is designed to detect transcriptional activation or repression, depending on whether effectors bind to the promoter (in this case, 5xUAS) to either activate or repress LUC gene expression. It is unclear how HDA19 functions as a repressor by itself in this system. Additionally, did the authors examine histone acetylation levels at the 5xUAS site? Does HDA19 repress LUC gene expression through histone deacetylation?

Response: Thanks for the suggestion. It is interesting that while HDIP1 plays a major role in mediating the repression of the *LUC* reporter gene, HDA19 by itself can weakly repress the expression of the *LUC* reporter gene (Fig 5I and Appendix Fig S13D). We predict that HDA19 may mediate transcriptional repression either through histone deacetylation or through certain unknown histone deacetylation-independent mechanisms. To validate this hypothesis, we performed ChIP-qPCR to determine whether histone deacetylation is involved in repressing the expression of the *LUC* reporter system. As shown in Fig5J, the H3Ac levels of the $5 \times UAS$ and *LUC* loci were not affected by different

effectors, even when the expression level of the reporter gene was significantly affected (Fig 5I and Appendix Fig S13D). These results strongly supports the notion that the HDIP-containing HDA19 complex can mediate transcriptional repression in a histone deacetylation-independent manner.

3. Fig 6. As in fig6I showed that mutation in HDIP1 cause variations in bolting time. Would mutations in HDIP1 affect its binding in vivo lead to the phenotype? Or influence the interaction with its co factors?

Response: Our phenotypic analysis indicated that the wild type *HDIP1-Flag* transgene but not the *HDIP1- Δ DBD-Flag* transgene can complement the defects of the *hdip1/2/3* mutant in the regulation of flowering time and silique development (Fig 6I and Appendix Fig S15A-C), suggesting that the DNA-binding ability of HDIP1 is required for its biological function in Arabidopsis plants. To investigate whether the DNA-binding ability of HDIP1 is involved in the genome-wide association of HDIP1 with chromatin, we performed the ChIP-seq analysis using wild-type HDIP1-Flag and HDIP1- Δ DBD-Flag transgenic plants and found that the deletion of the DNA-binding domain (Δ DBD) did not affect the genome-wide association of HDIP1 with chromatin (Appendix Fig S15D,S15E), suggesting that the DNA-binding domain of HDIP1 is not required for the association of HDIP1 with chromatin. Moreover, our AP-MS analysis showed that the deletion of the DNA-binding domain (Δ DBD) did not disrupt the incorporation of HDIP1 into the HDA19-containing HDAC complex (Appendix Fig. S15F). Additionally, our ChIP-seq data indicated that the *hdip1/2/3* mutation did not affect the association of HDA19 with chromatin at the whole-genome level (Appendix Fig S11A-S11C), suggesting that the DNA-binding capacity of HDIP1 is not involved in the association of HDA19 histone deacetylase complex with chromatin. These results suggest that the DNA-binding domain of HDIP1 takes effect after the HDA19 complex is associated with chromatin. Given that the DNA-binding domain is crucial for the biological function of HDIP1, we predict that the DNA-binding ability of HDIP1 is crucial for facilitating HDA19-mediated histone deacetylation in the HDA19 complex.

Minor: Fig7. It is unclear why figure 7 is necessary for this manuscript.

Response: Previous studies indicated that conserved components of histone deacetylase complexes are involved in ABA and stress response (Mehdi et al, 2016; Perrella et al, 2013). The results shown in Fig 7 suggest that HDIPs and HDA19 can co-regulate ABA sensitivity and stress tolerance, further confirming that HDIPs can function as components of the HDA19 complex in Arabidopsis plants. Moreover, Fig 7 indicated that the HDIP1/2/3 can HDA19 can suppress the expression of several stress-responsive genes in a histone deacetylation-independent manner, reinforcing the conclusion of this study. Therefore, we include Fig 7 in the main text of the study.

References

- Mehdi S, Derkacheva M, Ramström M, Kralemann L, Bergquist J & Hennig L (2016) The WD40 Domain Protein MSI1 Functions in a Histone Deacetylase Complex to Fine-Tune Abscisic Acid Signaling. *Plant Cell* 28: 42–54
- Perrella G, Lopez-Vernaza MA, Carr C, Sani E, Gosselé V, Verduyn C, Kellermeier F, Hannah MA & Amtmann A (2013) Histone deacetylase complex1 expression level titrates plant growth and abscisic acid sensitivity in Arabidopsis. *Plant Cell* 25: 3491–3505

Dear Xin-Jian,

Thank you submitting a revised version of your manuscript. It was sent to the same three reviewers that originally appraised your work; their comments are attached to the bottom of this email. As you will see, all three referees are satisfied with the changes you made. Before we can move forwards towards publication of your manuscript, there are some remaining editorial points which need to be addressed. In this regard, would you please:

- remove the author credit section from the manuscript,
 - include text callouts in the manuscript for Fig. 2A-G,
 - remove Dataset legends from the manuscript and upload them as a separate tab/sheet in each relevant Excel file,
 - remove Appendix figure legends from the manuscript, include a title page with "Appendix on Essential angiosperm-specific subunits of HDA19 histone deacetylase complexes in Arabidopsis" and a table of contents with page numbers of listed items missing, use nomenclature 'Appendix Figure Sx' throughout the Appendix PDF,
 - include a 'Resources and Tools' table uploaded as an individual file using the template from our guide to authors,
 - ensure source data files for Fig. 1A and 1F are deposited in an external repository (PRIDE is suggested) and fill in the relevant information in the source data checklist (a blank copy has been uploaded for your convenience),
 - remove reviewer access codes for the manuscript and ensure all data archived in public repositories are fully and publicly available,
 - provide exact p values in the legends of figures 3B and 4D
- indicate the statistical test used for data analysis in the legends of figures 3B, D; 4D and 5D,
- define box plots in terms of minima, maxima, centre, bounds of box and whiskers, and percentile in the legends of figures 4F, 5E and F,
 - define n in the legends of figures 4F, 5E and F,
 - define error bars in the legends of figures 7B, D, E,
 - rename the 'Materials and methods' as 'Methods', and
 - correct the section order as follows: Title page - Abstract & Keywords - Introduction - Results - Discussion - Methods - Data Availability - Acknowledgements - Disclosure and Competing Interests Statement - References - Figure Legends - Table(s) - Expanded View Figure Legends.

Please also note that (should you so wish) we are able to include author names using Chinese characters (for an example, see <https://www.embopress.org/doi/epdf/10.1038/s44318-024-00147-9>).

We include a synopsis of the paper (see <http://emboj.embopress.org/>). Please provide me with a two-sentence general summary statement and 3-5 bullet points that capture the key findings of the paper.

We also need a summary figure for the synopsis. The size should be 550 wide by [200-400] high (pixels). You can also use something from the figures if that is easier.

I look forward to receiving these changes. EMBO Press is an editorially independent publishing platform for the development of EMBO scientific publications.

Best wishes,

William

William Teale, PhD
Editor
The EMBO Journal
w.teale@embojournal.org

- a point-by-point response to the referees' comments, with a detailed description of the changes made (as a word file).
 - a word file of the manuscript text.
 - individual production quality figure files (one file per figure)
 - a complete author checklist, which you can download from our author guidelines (<https://www.embopress.org/page/journal/14602075/authorguide>).
 - Expanded View files (replacing Supplementary Information)
- Please see out instructions to authors
<https://www.embopress.org/page/journal/14602075/authorguide#expandedview>
- a Reagents and Tools Table as part of the Methods section, which can be downloaded from our author guidelines (<https://www.embopress.org/page/journal/14602075/authorguide#structuredmethods>)

We realize that it is difficult to revise to a specific deadline. In the interest of protecting the conceptual advance provided by the work, we recommend a revision within 3 months (6th May 2025). Please discuss the revision progress ahead of this time with the editor if you require more time to complete the revisions. Use the link below to submit your revision:

Referee #1:

The authors have satisfactorily addressed most of my concerns.

Referee #2:

The revised manuscript by Liu et al. has successfully addressed all my concerns and requests. This is a highly interesting and thorough study on the biology of histone deacetylation complexes in plants, offering valuable insights that will be of broad interest to the chromatin and gene expression communities. I congratulate the authors on an excellent manuscript and look forward to its publication.

Referee #3:

The author has done vigorous revision and have addressed all my concerns. I have no further questions.

All editorial and formatting issues were resolved by the authors.

Dear Dr. He,

I am pleased to inform you that your manuscript has been accepted for publication in the EMBO Journal.

Congratulations!

Yours sincerely,

William Teale

William Teale, PhD
Editor
The EMBO Journal
w.teale@embojournal.org
